# Perturbation of amygdala-cortical projections reduces ensemble coherence of palatability coding in gustatory cortex

Jian-You Lin[1,2], Narendra Mukherjee[2], Max J Bernstein[1,2], Donald B Katz[1,2]*

[1]Department of Psychology, Waltham, United States; [2]The Volen National Center for Complex Systems, Brandeis University, Waltham, United States

**Abstract** Taste palatability is centrally involved in consumption decisions—we ingest foods that taste good and reject those that don't. Gustatory cortex (GC) and basolateral amygdala (BLA) almost certainly work together to mediate palatability-driven behavior, but the precise nature of their interplay during taste decision-making is still unknown. To probe this issue, we discretely perturbed (with optogenetics) activity in rats' BLA→GC axons during taste deliveries. This perturbation strongly altered GC taste responses, but while the perturbation itself was tonic (2.5 s), the alterations were not—changes preferentially aligned with the onset times of previously-described taste response epochs, and reduced evidence of palatability-related activity in the 'late-epoch' of the responses without reducing the amount of taste identity information available in the 'middle epoch.' Finally, BLA→GC perturbations changed behavior-linked taste response dynamics themselves, distinctively diminishing the abruptness of ensemble transitions into the late epoch. These results suggest that BLA 'organizes' behavior-related GC taste dynamics.

*For correspondence: dbkatz@brandeis.edu

**Competing interests:** The authors declare that no competing interests exist.

## Introduction

A significant part of our daily lives is spent on acquiring and consuming food and drink. The ultimate goal of this pursuit is the ingestion of nutrients that satisfy bodily needs and maintain physiological health, but our food choices are seldom consciously made to satisfy such needs. Rather, we eat food that is delicious, regardless of whether it is nutritious (*Baldo et al., 2016*; *Marshall et al., 2017*). In many situations, this still poorly understood (*Berridge, 2000*) drive to consume high-palatability food overwhelms and subverts the need for nutrition (e.g. binge eating; *Yeomans et al., 2004*).

Given the centrality of palatability to consumption decisions, it is unsurprising that palatability-related activity is prominent in gustatory cortical (GC) taste response dynamics. Across the 0.2–1.5 s period following taste delivery (the initial 0.2 s of responses is non-specific), GC neural ensembles progress through a sequence of processing 'epochs,' such that, after briefly coding chemosensory information (the taste 'Identity Epoch'), responses become dominated by activity correlated with hedonics (the 'Palatability Epoch;' *Katz et al., 2001*; *Katz et al., 2002*; *Fontanini and Katz, 2006*; *Sadacca et al., 2012*; *Maier and Katz, 2013*; *Sadacca et al., 2016*). The transition between these epochs occurs suddenly and coherently across GC, a fact that can be observed using ensemble analyses such as Hidden Markov Modeling (HMM; *Jones et al., 2007*; *Sadacca et al., 2016*). These analyses make it possible, despite significant trial-to-trial variability, to accurately identify the onset latency of the Palatability Epoch in single trials, and thereby to show that this ensemble event accurately predicts the onset of palatability-specific orofacial responses (e.g. gaping, an egestive response typically evoked by aversive bitter taste stimuli; *Li et al., 2016*; *Sadacca et al., 2016*). Furthermore, perturbing the neural activity preceding this transition interferes with production of these palatability-driven behaviors (*Li et al., 2016*; *Mukherjee et al., 2019*).

The very nature of GC taste response dynamics themselves—their complexity, their coherence, and the transitions in functionality—implies the functioning of complex networks, suggesting that GC does not perform this task alone (*Jones et al., 2006*). In fact, GC does receive potentially relevant input from several brain areas (*Krettek and Price, 1977*; *Saper, 1982*; *Allen et al., 1991*), notably including the basolateral amygdala (BLA), a region that: (1) is reciprocally connected with GC (*Stone et al., 2011*); (2) codes taste palatability (*Fontanini et al., 2009*); and (3) participates generally in reward-guided behavior (*Nishijo et al., 1998*; *Blundell et al., 2001*; *Balleine et al., 2003*; *Holland and Gallagher, 2004*). As palatability-related information emerges one epoch earlier in BLA than in GC (*Fontanini et al., 2009*), it could be suggested that taste hedonics are 'passed' between the two. Support for this specific hypothesis has come from a study showing that pharmacological inactivation of BLA impacts GC taste coding (*Piette et al., 2012*).

The interpretability of this earlier study is limited by several factors, however. First, whole-region pharmacological inhibition impacts all projection pathways; the effect of BLA inhibition on GC taste coding could be wildly indirect, involving (among others) brain regions such as lateral hypothalamus (*Krettek and Price, 1978*; *Saper et al., 1979*; *Berk and Finkelstein, 1982*; *Petrovich et al., 2001*; *Berthoud and Münzberg, 2011*) and/or the parabrachial nuclei in the pons (*Lundy and Norgren, 2004*; *Li et al., 2005*), both of which also code taste palatability (*Li et al., 2013*; *Baez-Santiago et al., 2016*). Furthermore, pharmacological inhibition persists for hours, a fact that introduces the possibility that circuit plasticity (rather than real-time inactivation) might explain the manipulation's effect on GC coding, and that also renders within-session comparisons of conditions impossible (greatly limiting the hypotheses that can be tested).

Here, we use pathway-specific optogenetics to directly test whether (and how) BLA input controls GC population coding of taste palatability. We discretely perturbed BLA→GC axons for 2.5 s starting at the time of taste delivery, without silencing somas in either structure. Our results demonstrate that this perturbation impacts GC taste responses in an 'epoch-wise' manner, in that: (1) the likelihood of firing-rate changes peaks at epoch onset times, despite the perturbation itself being tonic; (2) the perturbation reduces Palatability Epoch content, without reducing Identity Epoch information; and (3) the loss of BLA input 'blurs' the onset of the Palatability Epoch by reducing the suddenness of the firing-rate transition in all neurons in the ensembles. These data suggest BLA to be more involved in the organization of emergent network dynamics than in the delivery of palatability information to GC per se.

## Results

### Perturbation of BLA→GC axons (BLA→GCx) impacts taste responses

We analyzed AAV-induced gene expression *via* immunohistochemical evaluation of the presence of GFP. A representative example of these data is shown in *Figure 1*. Note the cell body staining in BLA, and the utter lack of cell body staining in GC, where expression is restricted to axon filaments. ArchT (which in this case co-expresses with GFP) is carried from the injection site in BLA in a purely anterograde direction, and a subset of infected axons terminate in the ventral part of GC—a result consistent with earlier rat and mouse data (*Haley et al., 2016*; *Levitan et al., 2019*). Rats in which GFP expression was not found in BLA and GC, or in which opto-trodes were misplaced, were excluded from further data analysis.

The data reported below were recorded from four rats in which *post-hoc* histological examination confirmed good electrode and fiber placements, and substantial virus expression; the dataset included a total of 140 neurons. For two recording sessions/rat (separated by one rest day), a battery of basic tastes (sucrose, NaCl, citric acid, and quinine HCl) were delivered via IOC. The impact of BLA→GCx on GC activity was analyzed using a 'within-subject' approach, whereby we compared GC neural responses in Laser-Off and Laser-On trials. Preliminary analyses revealed that neither the percentage of recoded neurons impacted by laser stimulation nor the direction of impact (suppression vs enhancement) significantly differed between the first and second sessions (all $X^2$ <1). Nor did strength of impact (*t*-values comparing control and perturbed trials) significantly differ between recording sessions ($F$ = 1.95, p>0.05). This pattern of results suggests that in the current experimental setting, the novelty of tastes (i.e., difference in taste familiarity between 1st and 2nd sessions)

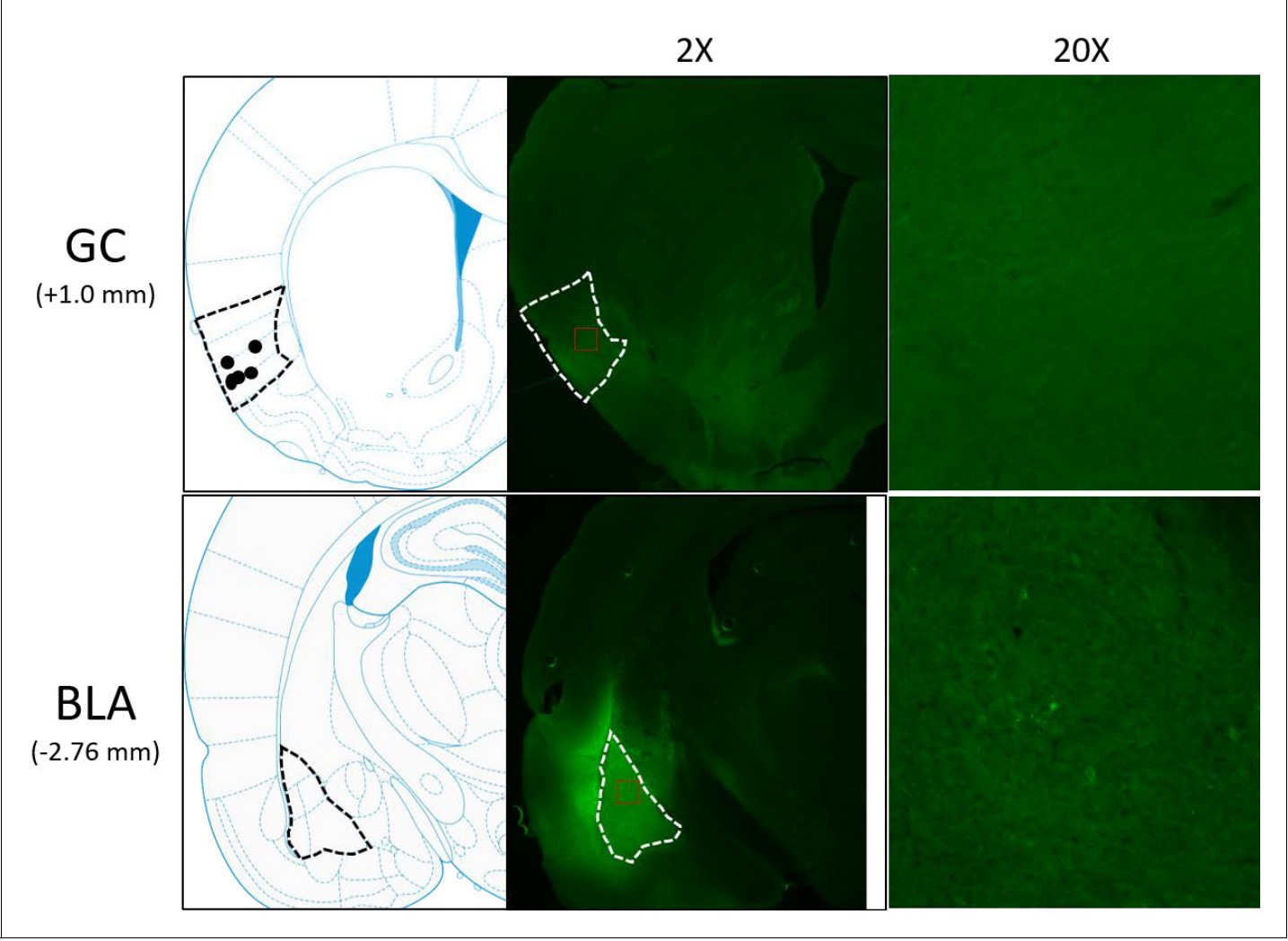

**Figure 1.** Schematics and sample histology showing ArchT virus infection, visualized by the GFP tag, in GC (top panel) and BLA (bottom panel). Brain slices were taken from 1.0 mm anterior to bregma for GC and 2.76 mm posterior to Bregma for BLA. Black and white dashed lines outline GC and BLA in each panel, and ×20 images were the magnification of areas within red dashed lines in ×2 images. Solid circles in the schematics are final locations of the tips of opto-trodes (Schematics were modified from *Paxinos and Watson, 2005* with permission will be requested from Elsevier upon the publication).

plays little role in the impact of BLA→GCx on GC taste response. Accordingly, data from the two sessions were pooled together, without inclusion of session as an additional analysis variable.

Activation of the optical silencer ArchT in BLA→GC axons ( i.e. laser illumination of GC) at the time of taste delivery had a strong impact on taste responses, but it was immediately clear that this impact wasn't a simple general reduction of taste response magnitude. *Figure 2* shows representative examples of the various ways in which BLA→GCx changed GC taste activity. Each panel shows raster plots (top) and the peri-stimulus time histogram (PSTH; bottom) of the taste response of a single GC neuron to taste presentation, with laser-on and laser-off trials plotted separately. For the purpose of visualization, the responses shown here were averaged across all taste trials (these examples were chosen on the basis of the impact of laser stimulation being comparable for all taste stimuli). In some cases, taste responses were unaffected by the laser (*Figure 2A*), whereas in others the responses were enhanced (*Figure 2B*) and in still others they were reduced in magnitude (*Figure 2C–D*).

To further investigate the effect of BLA→GCx on the activity of individual GC neurons, the differences between laser-on and laser-off responses were calculated for ten 250 ms time bins post taste delivery. *Figure 2E* summarizes this analysis, showing that the impact of BLA→GCx on individual GC

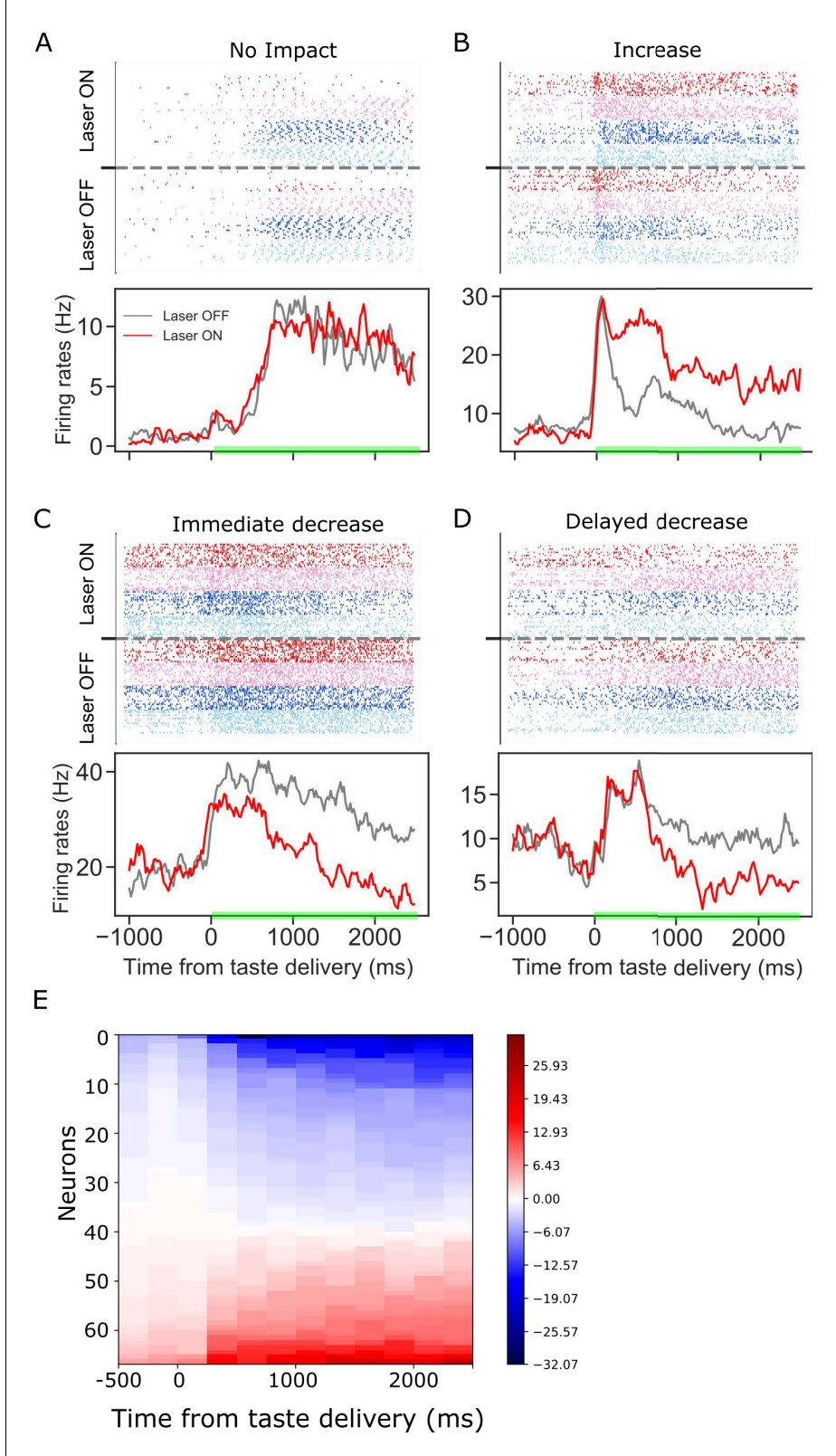

**Figure 2.** Panels A-D: GC taste responses with raster plots (top) and peri-stimulus histogram (bottom; averaged across all taste trials) during Laser-Off and Laser-On trials. Each panel shows a representative GC neuron whose activity: (A) was not modulated, (B) increased, (C) immediately decreased, or (D) had a delayed decrease due to BLA→GCx. (E). Mean firing rate differences between control and perturbed trials across 2.5 s (x-axis, divided into

*Figure 2 continued on next page*

*Figure 2 continued*

ten 250 ms-bins) post-taste delivery. Each row in the y-axis is an individual neuron for which activity was significantly altered by BLA→GCx. Blue and red colors indicate the degree to which responses were decreased or increased by the perturbation, respectively. In each case, the perturbation either deceased or increased GC activity, but not both.

---

neurons was reliable in direction (increase or decrease of firing rates) across the entire duration of the change, as indicated by the relative lack of switching between blue and red colors within any individual row (i.e. neuron) of the heatmap. Even with sub-threshold (i.e. non-significant [$ts < 1$]) changes included in the plot, positive and negative firing-rate changes are found in the same neuron only once. The influence of BLA on GC activity can be to increase or decrease firing, but is largely unimodal for individual neurons.

Furthermore, the response changes wrought by the laser did not simply reflect the laser-on time: the initial 150–200 ms of the responses (i.e. the period preceding the two taste coding epochs) was unaffected by the laser (only ~10% of our GC neurons were affected in this early period by the perturbation, also see Figures 5C and 6A), and in some cases the latency to the laser's impact was substantially longer (e.g. *Figure 2D*). The dispersion of these latencies appears, at least visually, to reflect the timing of the epochs making up the dynamics of GC taste responses (*Katz et al., 2001*; *Fontanini and Katz, 2006*; *Jones et al., 2007*; *Grossman et al., 2008*; *Miller and Katz, 2010*; *Sadacca et al., 2012*; *Maier and Katz, 2013*; *Sadacca et al., 2016*).

Below, we unpack and test these observed impacts of BLA→GCx on taste response firing in whole sample analyses—first examining the magnitudes and directions of the firing rate changes, and then the epoch-specific nature of the changes.

## Both enhancement and suppression of GC taste response firing are wrought by optogenetic perturbation of BLA→GC axons

A total of 55.7% of the recorded GC neurons (78 out of 140) produced taste responses that were impacted by BLA→GCx (*Figure 3A*: pie chart on the left). Herein, we defined a neuron as impacted by laser stimulation if one or more of its control-trial taste-evoked responses (NaCl, Sucrose, Acid, or QHCl) were significantly different from those in perturbed trials. The modal result was broad changes—in 46.1% of the neurons affected, the perturbation changed responses to all four taste stimuli, although in other above-chance fractions of recorded neurons BLA→GCx altered responses to fewer tastants (*Figure 3B*). In total, ~40% of the individual taste responses were impacted by the perturbation (*Figure 3C*).

In cases in which BLA→GCx impacted >1 taste response (63 out of 78 neurons; the solid portion of *Figure 3D*), the perturbation either consistently suppressed (57.1%: 36 out of 63) or consistently enhanced (42.9%: 27 out of 63) response magnitudes for all affected taste responses (*Figure 3D*). That is, if a given neuron's activity was significantly altered by BLA→GCx, the direction of impact (suppression/enhancement) was the same across all taste responses.

Given that in vitro slice recordings e.g. *Haley et al., 2016*; *Haley et al., 2020* have shown BLA projection neurons to synapse onto both pyramidal cells (PCs) and interneurons (INs), it was important to ask whether the effect of BLA→GCx was cell-type dependent. Our in vivo electrophysiology does not permit definitive determination of cell type, but we were able to distinguish putative PCs from putative INs based on the shape of the spike waveforms (*cf. Sirota et al., 2008*; *Quirk et al., 2009*; *Herzog et al., 2019*). The ease with which neuron groups could be distinguished using this criterion is shown in *Figure 4A*, along with representative neurons of each type (note the difference in the 2nd half width of the two action potentials). Consistent with a great deal of prior research (e.g. *Quirk et al., 2009*), putative INs showed (on average) significantly higher basal firing rates than did putative PCs (*Figure 4B*).

After dividing the sample on this basis, we were in fact able to observe clear differences in the impact of BLA→GCx on putative PCs and INs. As shown in *Figure 4C*, laser stimulation suppressed taste responses in approximately half of the neurons identified as PCs (n = 21 [47%]), but among putative INs, the impact was almost always suppression (n = 19/21 [90%]). This result is consistent with data (e.g. *Saper, 1982*; *Allen et al., 1991*) suggesting that BLA→GC projection neurons provide excitatory glutamatergic input to both PCs and INs: it would be expected that some changes in

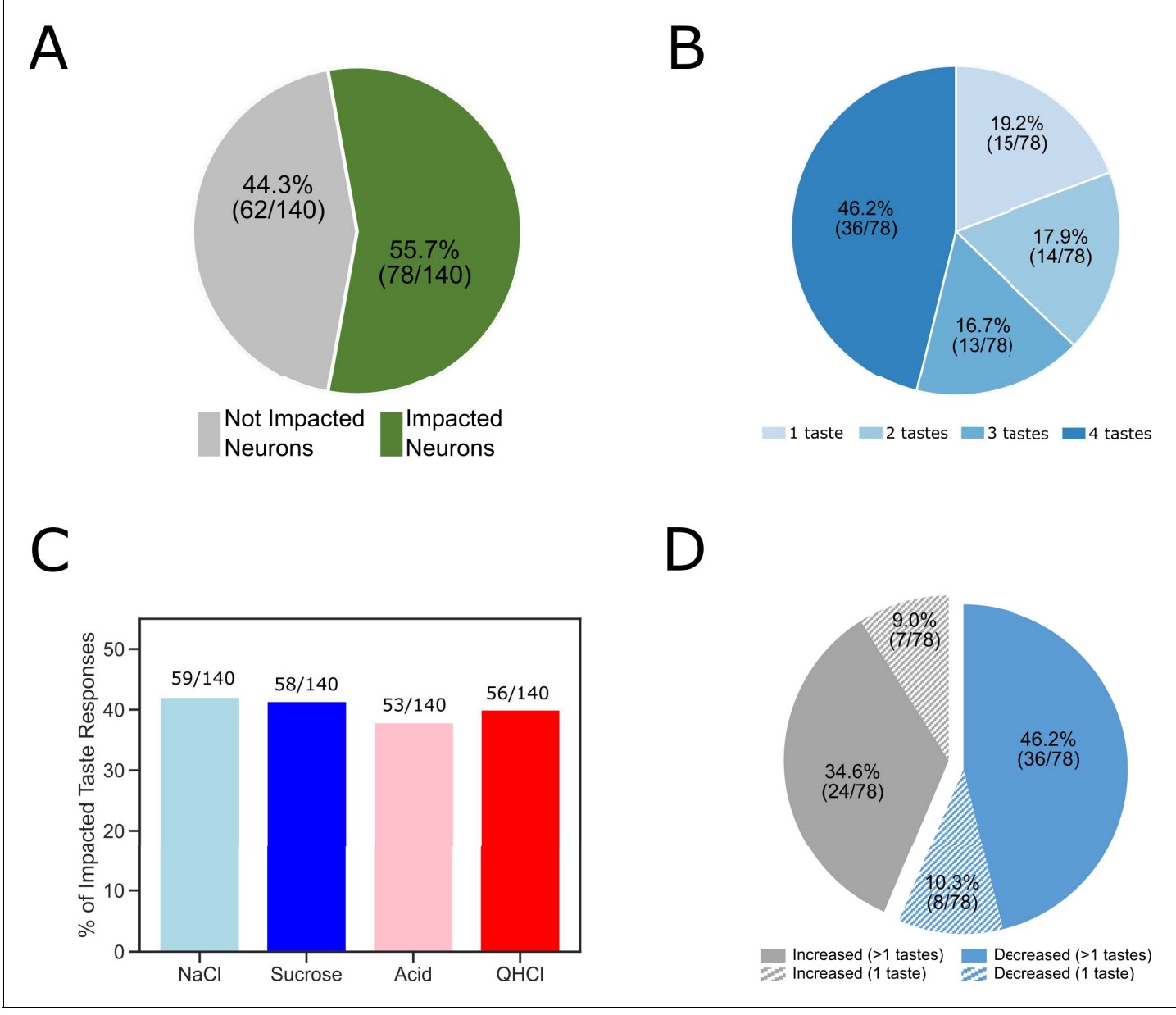

**Figure 3.** The overall impact of perturbation of BLA→GC axons on the entire GC population. (A) Pie chart showing 55.7% (78/140) of recorded GC neurons were affected by laser stimulation. (B) The impacted GC neurons with various taste responses being influenced by laser stimulation. For 46.2% of the impacted neurons, responses to all four tastes were changed; 16.7% of the neurons showed changes to three (out of 4) tastes; for 17.9% of the GC neurons changed their responses to two tastes. Finally, for the rest of the neurons there was only one taste response altered by perturbation. (C) Bar graph showing the number of each individual taste response being altered by BLA→GCx. As revealed, the impact was comparable across tastes:~40% of each taste response was altered by stimulation. (D) Among the impacted neurons, 56.4% (44/78; blue) of them decreased their response rates in reaction to taste delivery while 43.6% (34/48; gray) increased response rates. Noted that in cases in which BLA→GCx impacted >1 taste response (60 [36 + 24] solid color), the perturbation either consistently decreased (60%; 36/60) or increased (40%; 24/60) firing rates.

PC responding came directly via loss of (excitatory) BLA input, and some as the indirect effect of a loss of input to inhibitory INs (**Stone et al., 2011**; **Haley et al., 2016**); the fact that only a small portion of INs showed enhanced firing rates, meanwhile, likely reflects the fact that INs were impacted mostly by direct loss of BLA input (because PCs in sensory cortices make fewer local connections; see **Zhang et al., 2014**; **Haley et al., 2016**). Beyond this basic property, however, we observed no significant neuron-type differences—BLA→GCx changed taste responsiveness, specificity, or palatability-related activity similarly for INs and PCs (all Chi-Squared >0.05), a result consistent with our

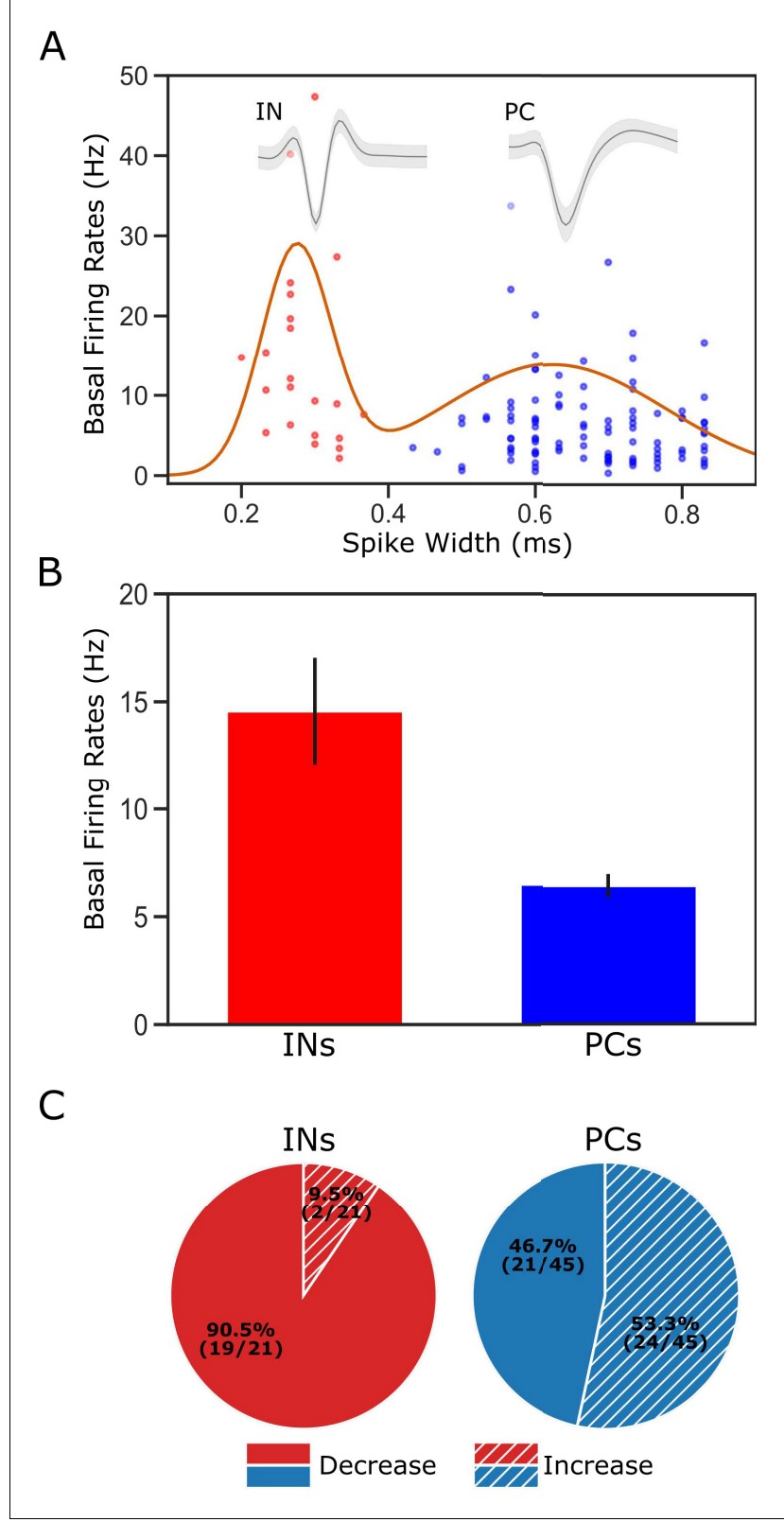

**Figure 4.** Classification of putative interneurons (INs) and pyramidal cells (PCs). (**A**) Given the radical difference in waveforms between INs and PCs (representative examples of each type shown on the top of the figure), neuronal type assignment was based on the half-spike width (calculated from valley to the post-valley peak); units with width <0.35 ms are marked as INs (red dots) while those with width >0.35 ms are classified as PCs (blue dots). A

*Figure 4 continued on next page*

*Figure 4 continued*

Gaussian fit to the firing rates of the recorded neurons is overlaid with the data points. (B) GC single units classified as INs have higher basal firing rates than those classified as PCs. (C) The impact of perturbation of BLA→GC axons was different among the two types of GC neurons. While most of the GC INs (90.5%) were suppressed, only 46.7% of PCs showed decreased firing rates following laser stimulation.

previous studies showing little evidence of neuron-type specificity in GC taste coding (*Katz et al., 2001*; *Fontanini and Katz, 2006*; *Jones et al., 2007*). Accordingly, we did not separate neurons into types for purposes of the subsequent analyses.

While these results suggest that perturbation of BLA→GC axons alters GC taste responses, it was important to consider the alternative possibility that the laser directly perturbs activity in GC neurons (despite the lack of obviously fluorescent GC somas). A comparison of our data with datasets previously collected in our lab (specifically, data collected as part of *Mukherjee et al., 2019*), however, allows us to reject this hypothesis, in that the impact of our manipulation is qualitatively different from direct optogenetic perturbation of GC neurons (*Figure 5A*). For one thing, our manipulation altered the taste responses of ~50% of recorded GC neurons (*Figure 3A*); direct activation of ArchT expressed in GC somas, meanwhile, changes the taste responses of almost all of the neurons (91% of the recorded population; *Figure 5B1*). The nature of the changes differs between the two preparations, as well: 87.9% of the response changes caused by direct optical perturbation of GC neurons involve suppression of firing (*Figure 5B2*)—a significantly different percentage than that caused by BLA→GCx in this experiment (60% in *Figure 3C*; $X^2$ = 17.07, p<0.01).

The effect of BLA→GCx is further differentiated from that of direct GC neuron perturbation with regard to the dynamics described above (*Figure 2*)—most notably, by the relatively late latency of BLA→GCx's impact on GC taste responses (*Figure 5C*; red sigmoid fit), and the accordingly late asymptotic effect (approximately 500 ms after taste delivery). When GC somas are themselves led to express ArchT channels, in contrast, laser illumination of GC causes immediate, steeply developing changes that reach asymptote within ~250 ms. We conclude that the changes in GC activity observed in the current experiment were not caused by perturbation of GC somas.

## Despite tonic laser illumination, the impact of BLA→GCx on GC firing rates is epoch-specific

As already noted, while the laser was turned on at the time of taste delivery, in the majority of cases optogenetic perturbation of BLA-GC axonal activity impacted GC taste response only after

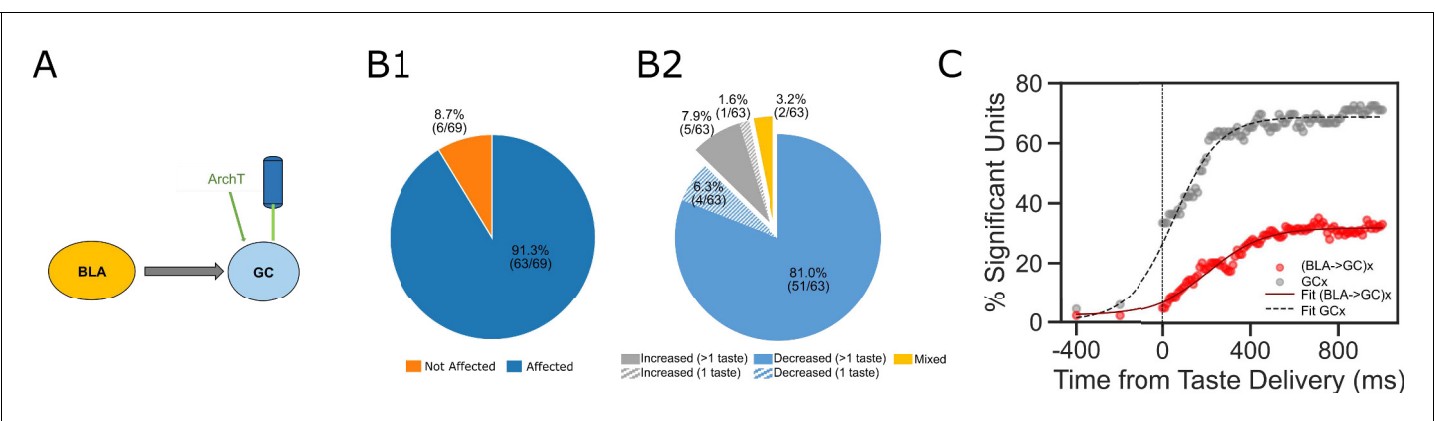

**Figure 5.** Comparisons between laser perturbation on GC somas and BLA→GC axons. (A) GC recording was done ~4 weeks after ArchT AAV virus was infused into GC. (B) Pie charts demonstrate that laser stimulation caused firing changes in 91.3% of the recorded GC neurons (B1), among which, 87.3% showed suppressed impact while others were either enhanced or showed mixed impact by the perturbation (B2). Noted that both the percentage of affected neurons and the percentage of decreased units following direct perturbation of GC somas are significantly greater than those with perturbation on BLA→GC axons (*Figure 3*). (C) Percentages of GC neurons affected by perturbation of GC somas (grey dots) or BLA→GC axons (red dots). As indicated by fitted lines (of sigmoid function); the percentages of impacted GC neurons not only started higher immediately following laser stimulation but also rose at a faster rate than that was found with BLA→GCx.

substantial delays, leaving the initial 200 ms of the responses unaltered (*Figures 2* and *5C*). In many cases, the latency of the effect was much longer. As an initial exploration of this phenomenon, we summarized the distribution of laser impact latencies across the entire sample of taste responses (*Figure 6A*; the red dashed line is the distribution smoothed with a Gaussian). Inspection of this panel reveals that the onset of changes caused by BLA→GCx is neither uniform nor a simple decay function (which would be the two most likely results if BLA input played no role in GC temporal coding). Instead, there are multiple peaks in the function, reflecting multiple 'most likely times' for the onset of perturbation-related changes. One such peak appeared between 300 and 350 ms after taste delivery, and a second appeared approximately 750 ms after taste delivery. A bi-modal fit of the data suggested the timing these peaks to be 347.28 (SD = 110) and 754.39 ms (SD = 179), and attempts to fit the data with an exponential decay function produced a lower coefficient of determination (an index of absolute goodness of a fit) than that of the Gaussian mixture model; moreover, the error scores at each time bin (i.e. estimated values – raw values) were significantly smaller with the bi-modal Gaussian fit than those with exponential fit ($t(24) = -2.39$, $p<0.05$).

This result, surprising given the tonic nature of the experimental manipulation, in fact dovetails remarkably well with 20 + years of research on the '3-epoch' dynamics of GC taste processing (*Katz et al., 2001*; *Jones et al., 2007*; *Sadacca et al., 2016*; *Mukherjee et al., 2019*; see also Discussion). Consistent with this observation, the nature of the perturbation's impact appears to shift around the time of the 2nd epoch (the time at which responses first become taste-specific, see *Katz et al., 2001*; *Fontanini and Katz, 2006*; *Sadacca et al., 2012*): response enhancements predominate prior to this point (*Figure 6B1*), while response suppressions are equally likely afterward (*Figure 6B2*); *Figure 6B3* summarizes this effect, showing the difference between the likelihood of firing-rate enhancements and suppressions. Together, these results suggest that inhibiting BLA input to GC across the first 2.5 s after taste delivery impacts taste processing in an 'epoch-wise' manner (keeping in mind that such measures are necessarily approximate, given the vagaries of detecting precise onsets of firing-rate reductions, see Discussion).

We next asked whether this 'epoch-wise' impact implied 'single-epoch' impact—whether firing rate changes with onset latencies around the time of the middle peak of *Figure 6A* might only last the length of the 'identity epoch,' ending around the beginning of the palatability epoch (i.e. around the time of the late peak in *Figure 6A*—the blue dashed arrow). *Figure 6C* disconfirms this possibility, showing that only 3 out of the 41 firing-rate modulations (i.e. 8%) were restricted to the identity epoch. There were, meanwhile, 15 neurons that were impacted only when the palatability epoch

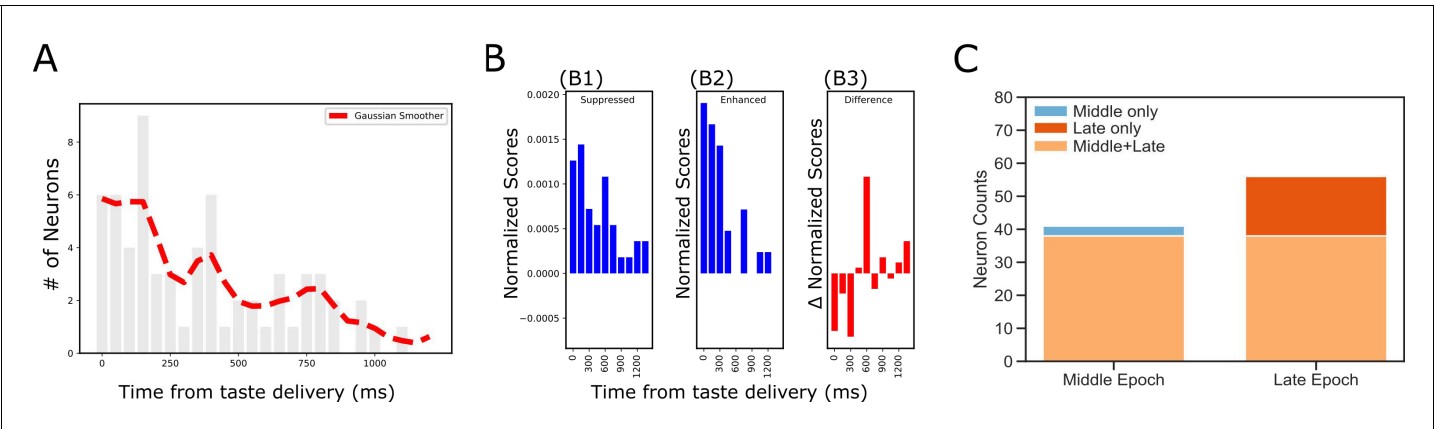

**Figure 6.** Dynamics revealed in analysis of impact latency and duration. (A) Histogram of impact latencies occurring post-taste delivery. Instead of an exponential decay function, the distribution is best fitted with a mixture Gaussian function (red dash line) which found two peaks at ~300 ms and ~750 ms, respectively. (B) Distribution of impact latency grouped by decreased impact (B1) and increased impact (B2). B3 reveals the difference between decreased and increased impact, demonstrating a clear sign shift in the time bin started at 600 ms; after the shift, decreased impact became more dominant than increased ones. (C) Number of laser-impacted GC neurons grouped by whether they showed significant laser impact during the Middle 'Identity' epoch (100–600 ms) or the Late 'Palatability' epoch (700–1200 ms). As revealed, BLA→GCx rarely only impacted Middle epoch firing (n = 3). Instead, most of the affected neurons showed significant impact over the late, palatability epoch with the impact that either started when the Late epoch began (n = 15) or started earlier and remained impacted over the Late epoch (n = 38).

began. This difference between epochs is significant ($X^2$ = 5.86, p<0.05), and it means that the majority of the effects of BLA→GCx are felt in the time period in which palatability-related processing is found.

In summary, 2.5 s perturbations of BLA input to GC change taste-driven activity in ways that are both non-random and complex—firing is modulated in specific relation to the dynamics that characterize GC taste processing. Such results imply, consistent with previous work (*Schoenbaum et al., 1998*; *Paré et al., 2002*; *Piette et al., 2012*), that disruptions of the BLA→GC pathway might have distinct consequences for different functional aspects of GC taste responses (aspects that have been shown to 'live' in the different response epochs; see *Sadacca et al., 2016*; *Mukherjee et al., 2019*); more speculatively, they imply that the dynamic nature of GC taste responses might itself be the product of interactions between the cortex and amygdala. Below, we test these two hypotheses.

## Perturbing BLA→GC axons selectively impacts (late epoch) palatability coding

Unlike taste responses in GC, those in BLA contain only two epochs, with the early detection epoch transitioning directly into the palatability-rich information epoch at ~200 ms following taste delivery (*Fontanini et al., 2009*). The lack of identity-related activity in BLA, the well-known involvement of BLA in value coding (e.g. *Johnson et al., 2009*; *Beyeler et al., 2016*; *Malvaez et al., 2019*), and epoch-specific laser impact on firing rates (*Figure 6*) together led us to hypothesize that: (1) taste discriminability would be at most only minimally altered by perturbation of BLA→GC axons—that our ability to identify the administered taste stimuli on the basis of GC single-neuron responses (and more particularly from responses in the middle, 'Identity' epoch) would survive the perturbation of BLA input, despite changes in absolute firing rates; and that (2) palatability processing, which is part and parcel of the Late epoch, would in contrast be greatly affected by BLA→GCx.

To assess the proposed (lack of) influence of BLA→GCx on GC identity coding, we first brought repeated-measures ANOVAs to bear on Identity epoch responses in laser and no-laser trials (separately), directly evaluating the incidence of taste specificity (i.e. whether a given neuron responded differently at least to one taste from other tastes across the first 2 s of taste processing) in each condition. While the perturbation did significantly change the firing rates of a large number (78 out of 140 in *Figure 3*) of neurons, we observed little evidence that BLA→GCx changed the incidence of taste-specific responses—the percentage of GC neurons responding in a taste-specific manner was identical (71.4%) in the two trial types (*Figure 7A*).

A closer look at this result revealed roughly similar distributions of individual taste responses in the two trial types (*Figure 7B*). Note that more than 50% of our GC sample responded to each taste; the fact that this percentage is far higher than 1/4 of 71.4% (the percentage of neurons that produced taste specific responses) means that GC neurons are broadly tuned—a result that is consistent with the vast majority of electrophysiological datasets involving >1–2 deliveries of each taste. GC neurons remained broadly tuned even when BLA→GC axons were perturbed via optogenetic inhibition, such that a chi-squared analysis failed to identify a significant difference between conditions (Laser Off vs. On; $X^2$ = 2.78, p>0.05).

Given the fact that BLA→GCx changed firing rates in GC taste responses (*Figures 2–6*), the results shown in *Figure 7A and B* imply that similar numbers of taste responses were created and destroyed by BLA→GCx. *Figure 7C* confirms this implication: several taste responses were lost when activity in BLA→GC axons was perturbed, but for each GC neuron for which taste specificity was lost, another neuron became a taste-specific neuron. Although BLA→GCx changed the specific composition of the neural ensembles producing taste-specific responses, the tastes continued to be coded by similarly sized GC populations when BLA input to GC was perturbed.

Of course, it remains possible that the magnitude of taste-specific information contained in the firing of each taste-responsive neuron was reduced by this manipulation—that despite there being similar numbers of taste-specific responses in both types of trials, the average 'magnitude' of taste specificity in the responses was reduced by the axonal perturbation. To evaluate this possibility, we asked how well tastes could be identified by these responses by subjecting the data from sets of simultaneously recorded GC neurons to a jack-knife classification test (*Foffani and Moxon, 2004*), testing the specific hypothesis that perturbation of BLA input reduces the distinctiveness (i.e. classifiability) of GC taste responses.

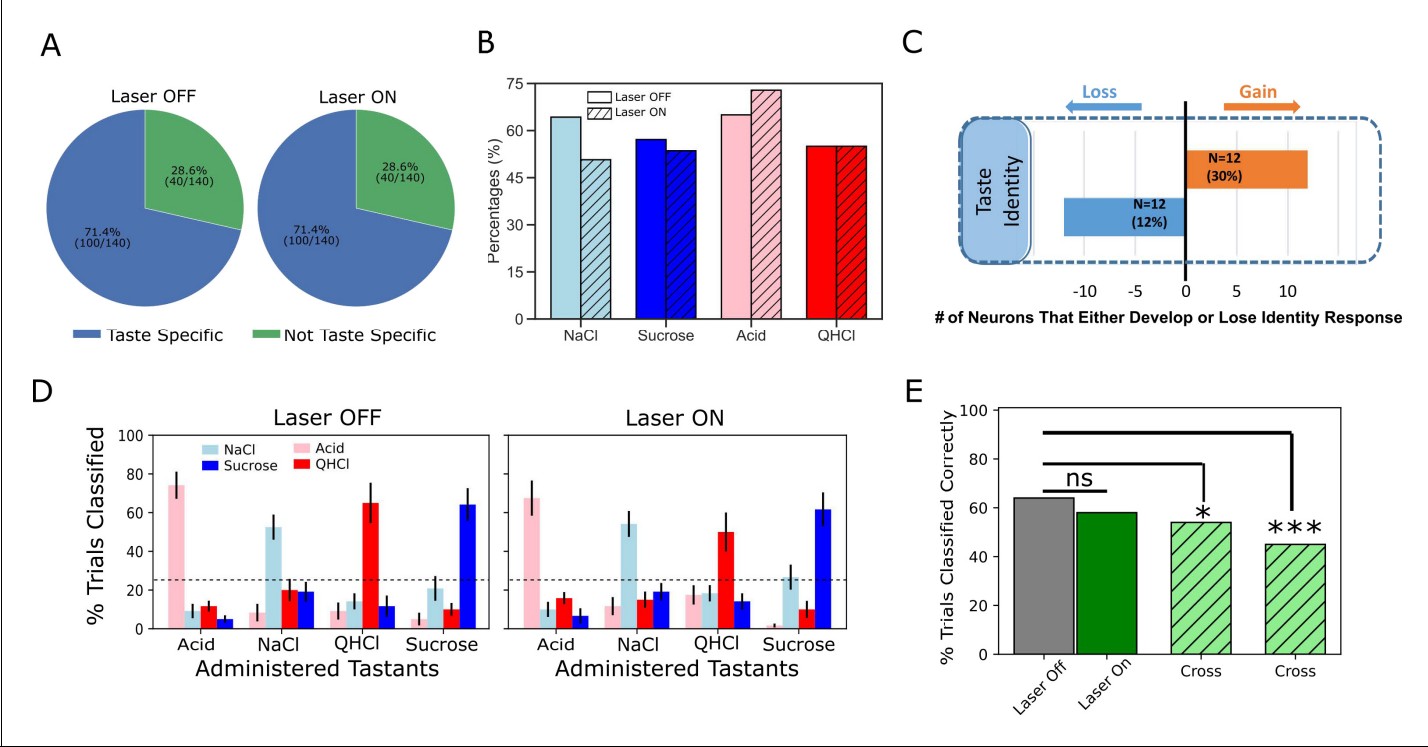

**Figure 7.** Laser perturbation of BLA→GC axons has little impact on GC taste specificity. (**A**) The numbers of taste-specific neurons (71.4%) in Laser Off and Laser On trials are essentially identical. (**B**) The proportions of neurons responding to each taste (NaCl, Sucrose, Acid, and QHCl) are similar (and statistically non-distinct) for Laser-Off and Laser-On trials. (**C**) A within-neuron analysis reveals that similar numbers of neurons lost taste-specific activity and developed taste-specific activity anew with laser stimulation. (**D**) A classification analysis performed on the entire GC dataset shows that discriminability of taste spiking responses was similar in Laser-Off (left panel) and Laser-On (right panel) trials, with the classifier reliably picking out the administered taste at well above chance levels (dashed lines indicate 25% chance performance). (**E**) The overall percentage of trials in which tastes were correctly identified is similar for Laser-Off and Laser-On trials. When classification was evaluated across laser conditions (i.e. when the classifier was trained on control trials and tested on laser trials or vice versa), the percentage of trials in which the taste was correctly identified dropped significantly ('Full dataset'; left hatched green bar); the rate of correct identification dropped still further when the analysis was restricted to neurons significantly impacted by laser (right hatched green bar). *<0.05, ***<0.001 in Chi-squared analyses.

As shown in *Figure 7D*, we failed to find substantial support for this hypothesis, in that BLA→GCx again proved to have little impact on the taste specificity of GC firing: the left panel confirms that GC single-neuron activity was reliably taste-specific—the classifier allowed us to correctly identify each administered taste (x-axis) on more than 50% of the held-out trials (y-axis), a percentage far higher than chance (25%). An essentially identical result was obtained from trials in which BLA input to GC was perturbed; furthermore, this held true regardless of whether we performed the analysis on whole-trial data or limited our analysis to firing within the Identity epoch.

The above result suggests that GC taste responses are discriminable in the absence of BLA input. This does not mean that the responses are unchanged by BLA→GCx, however; in fact, many Identity epoch responses were clearly changed by the input perturbation. To directly determine whether GC uses the same or different taste codes across laser conditions, we tested whether a classifier trained on one laser condition could be used to predict taste trials obtained from the other condition. The result of this analysis is displayed in *Figure 7E*; in this case, each bar represents the overall percentage of trials correctly predicted across different training/testing conditions.

The results of this analysis are plain: when the classifier was trained and tested on trials within the same laser condition (the solid gray and green bars), 64% of control trials and 58% of perturbed trials were correctly classified—the patterns of performance, both well above chance, do not differ from one another, demonstrating that BLA→GCx had no deleterious impact on the quality of coding content ($X^2 = 2.96$, p>0.05). Classification performance dropped, however, when the classifier was tested and trained on different trial types (left hatched green bar; $X^2 = 5.22$, p<0.05), a reduction

that became significantly worse (45% correctness; right hatched green bar) when the analysis was focused on neurons for which firing rates were changed by the input perturbation ($X^2 = 30.22$, $p<0.001$). Thus, while GC remains taste discriminative without BLA input, coding for taste identity is altered.

But this result, whereby inhibition of BLA→GC axons changes firing without having measurable impact on the magnitude of GC taste coding in the first ~750 ms of taste responses, contrasts strongly with the result of this same perturbation on Late-epochal palatability-related activity. The

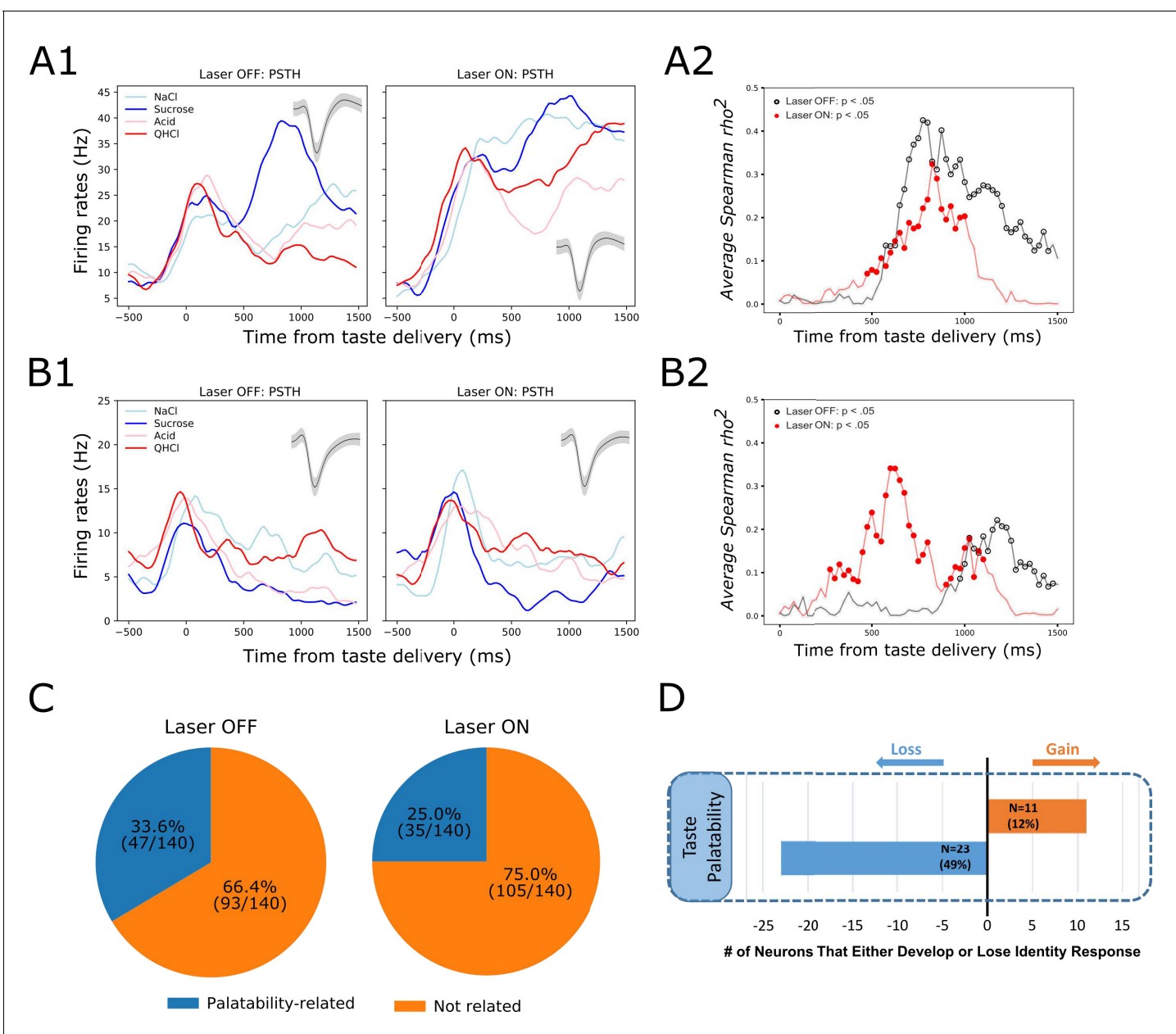

**Figure 8.** GC palatability information lost following perturbation on BLA→GC axons. (**A**) Representative GC neuron showing decreased palatability-related activity by stimulation. PSTHs of this neuron (Panel A1) were plotted over Laser-Off trials (left panel) and Laser-On trials (right panel). Panel A2 is the moving window analysis of Spearman correlation between firing rates and taste palatability across post-stimulus time. (**B**) Representative GC neuron showing increased (unmasked) palatability-related activity following laser stimulation; PSTHs of the neuron are plotted in Panel B1 and palatability correlation is shown in Panel B2. (**C**) Overall, the number of GC neurons showing palatability activity was significantly decreased by laser stimulation from 33.6% to 25.0%. (**D**) Within-neuron analysis revealed that whereas ~50% of the GC neurons (N = 23) that initially showed palatability activity lost the response during perturbation, only 12% of the neurons (N = 11) gained palatability responsiveness following stimulation.

representative example shown in *Figure 8A* illustrates this impact: *Figure 8A1* shows changes in Late-epoch taste responses, and *Figure 8A2* shows the attendant attenuation of palatability-relatedness in the normal pattern of firing (which is sucrose >NaCl > Acid>QHCl); the growth in firing-palatability correlation across the 2nd half-second after taste delivery in control trials (black open circle) is standard for GC taste responses, but in BLA→GCx trials this correlation rose more slowly (red circles in *Figure 8A2*), reached a lower asymptote, and disappeared more quickly.

While in some cases BLA→GCx actually increased palatability-relatedness of GC neurons (see the example in *Figure 8B1-2*), overall the number of neurons for which Late-epoch taste response firing rates were significantly correlated with palatability decreased with the perturbation (*Figure 8C*; between-condition $X^2$ = 5.49, p<0.05). *Figure 8D* reveals further details, showing that a far larger number (and percentage) of neurons lost palatability-related firing with BLA→GCx than gained (compare *Figure 8C & D* to *Figure 7A & C*). This result was corroborated by a direct comparison of the correlation between firing and palatability, which was lower—for Late epoch firing only—in perturbed trials (time x condition $F(1,139)$ = 5.70, p<0.05). Clearly, there was an overall loss of palatability-related firing in GC, in the absence of significant loss of identity-related information, when input from BLA was perturbed.

## Perturbation of BLA input to GC attenuates the ensemble properties of GC taste activity

The above single-neuron analyses support our hypothesis that direct inputs to GC from BLA are involved in GC palatability processing, but they also make it clear that this involvement is far from the whole story. Palatability-relatedness in the firing of some single neurons was not utterly eliminated by our input manipulation; in some cases it was even enhanced. This fact is perhaps somewhat surprising given the well-known importance of amygdala for emotion processing (e.g. *Quirk et al., 1995*; *Schoenbaum et al., 1998*; *LeDoux, 2000*; *Wang et al., 2005*; *Wassum and Izquierdo, 2015*; *Beyeler et al., 2018*), and findings suggesting that BLA-GC circuitry is vital for palatability-related behavior (CTA learning and taste neophobia; *Gallo et al., 1992*; *Lin and Reilly, 2012*; *Levitan et al., 2020*). Our recent data suggest a possible explanation, however: as previously discussed, the emergence of Late-epoch palatability coding is revealed, using single-trial analyses involving Hidden Markov Modeling (HMM), to be a sudden transition into a new ensemble state, in which firing-rate changes occur simultaneously in multiple GC neurons (*Jones et al., 2007*; *Miller and Katz, 2010*; *Sadacca et al., 2016*); it is this sudden transition itself that directly drives behavior (*Mukherjee et al., 2019*). Perhaps, the true extent of the perturbation effect is best apprehended, not in terms of changes in the magnitudes of palatability coding, but in terms of the ensemble coherence and/or suddenness of the transition into palatability-related firing.

To examine whether this might be the case—whether BLA→GCx alters the ensemble properties of this state transition—we subjected our data to Hidden Markov Modeling (HMM). *Figure 9A* shows 4 (consecutively collected control) example trials of spiking activity (vertical hash marks) in a set of simultaneously recorded neurons responding to (in this case) NaCl administration, with HMM-calculated probabilities (y-axis) of states, defined in terms of sets of firing rates across neurons, overlain (colored solid lines). As we have observed previously, the ensemble firing-rate transitions (the most likely times of state changes) occurred suddenly in control trials, reflecting the simultaneous precipitous changes in firing rates in multiple neurons, but varied in latency from trial to trial (e.g. *Jones et al., 2007*; *Moran and Katz, 2014*; *Sadacca et al., 2016*; *Mukherjee et al., 2019*). When these control trials were aligned to the onset of the state that occupied most of the duration between 500 and 1500 ms post-taste delivery (the period in which palatability-related firing emerges; see *Katz et al., 2001*; *Jones et al., 2007*; *Sadacca et al., 2012* and *Figure 8A2*) the sharpness of that transition into palatability-related firing was revealed (*Figure 9B1 and 2*, black dashed line; *Sadacca et al., 2016*; *Mukherjee et al., 2019*)—sharpness that is obscured in typical across-trial analyses.

When we brought the same analysis routine to bear in trials in which activity in BLA→GC axons was perturbed (*Figure 9B1*, dashed red line), the transition into palatability-correlated firing was far less steep than that in control trials from the very same sessions (and same neural ensembles). To quantify this finding for statistical evaluation, we fitted sigmoid curves to each transition function (black and red solid lines), and found that the slope of the rise into palatability-correlatedness was significantly lower for trials in which the laser was turned on than for trials in which the laser was off

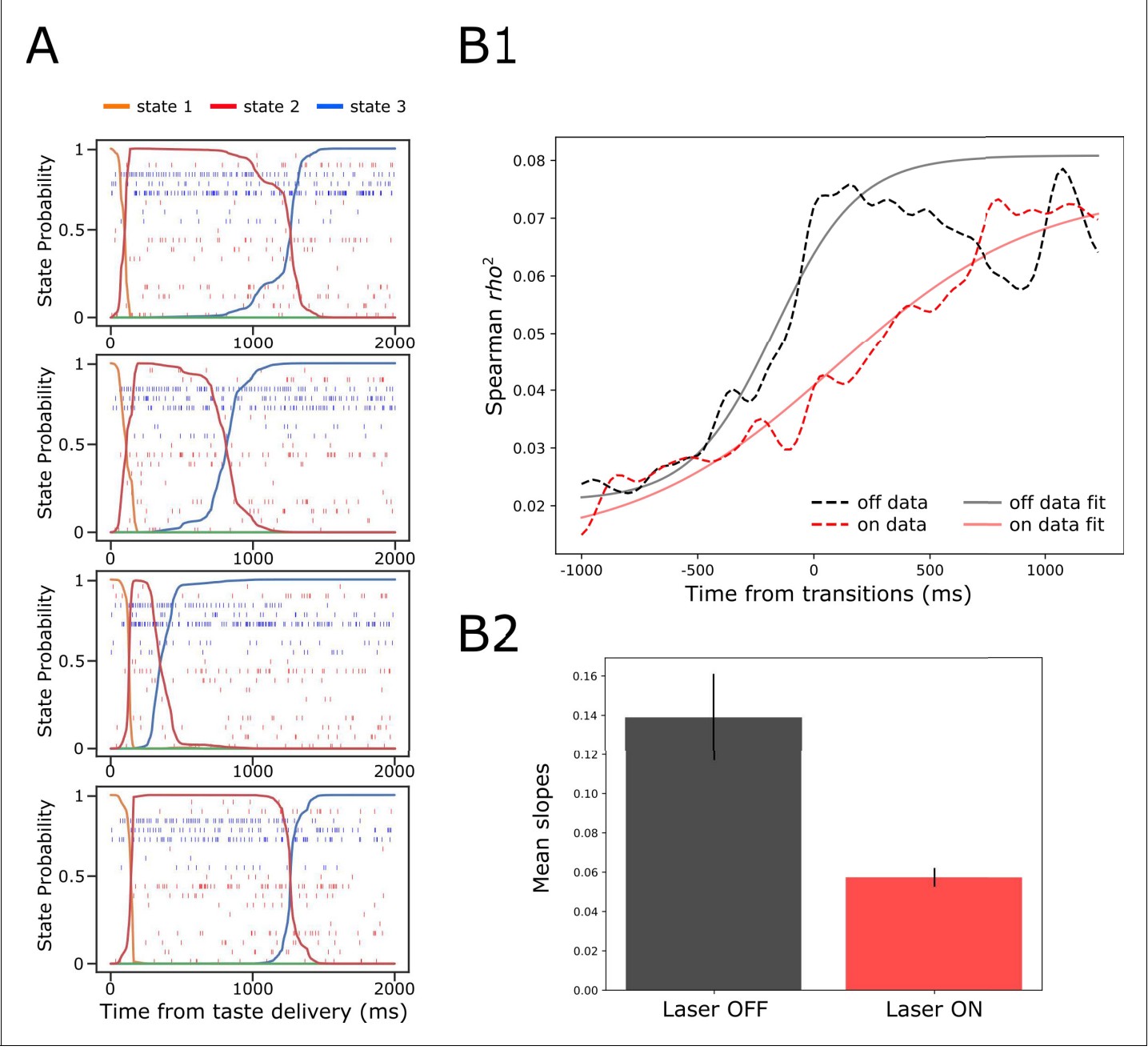

**Figure 9.** GC ensemble palatability activity was greatly impaired by perturbation on BLA→GC axons. (**A**) Example ensemble responses evoked by NaCl administration characterized using HMM: The colored lines overlain on the ensemble of spike trains (each row representing a single neuron, y-axis) indicate the calculated probability that the ensemble is in that particular state. (**B1**) Solid lines are moving window analysis of palatability correlations between firing rates (calculated from spike trains ranged from 1000 ms before to 1500 ms after the transition time) and taste palatability; dashed lines are sigmoid fits for the raw data. (**B2**) The slopes of the sigmoid fits in B1 (error bars denote 95% Bayesian credible intervals); the development of correlations is significantly shallower on the perturbed (Laser-On) trials than on control (Laser-Off) trials.

(as indicated by the lack of overlap between the 95% credible intervals in *Figure 9B2*; see Materials and methods). This result makes it clear that, with activity in BLA→GC axons perturbed, GC ensemble taste activity fails to transition with normal suddenness into palatability-related firing.

We considered two possible explanations for this result (see *Figure 10A*): (1) the possibility that perturbation of BLA input during taste responses caused a general reduction in the sharpness of firing-rate changes for all individual neurons in an ensemble (Figure 10A-top); and (2) the possibility

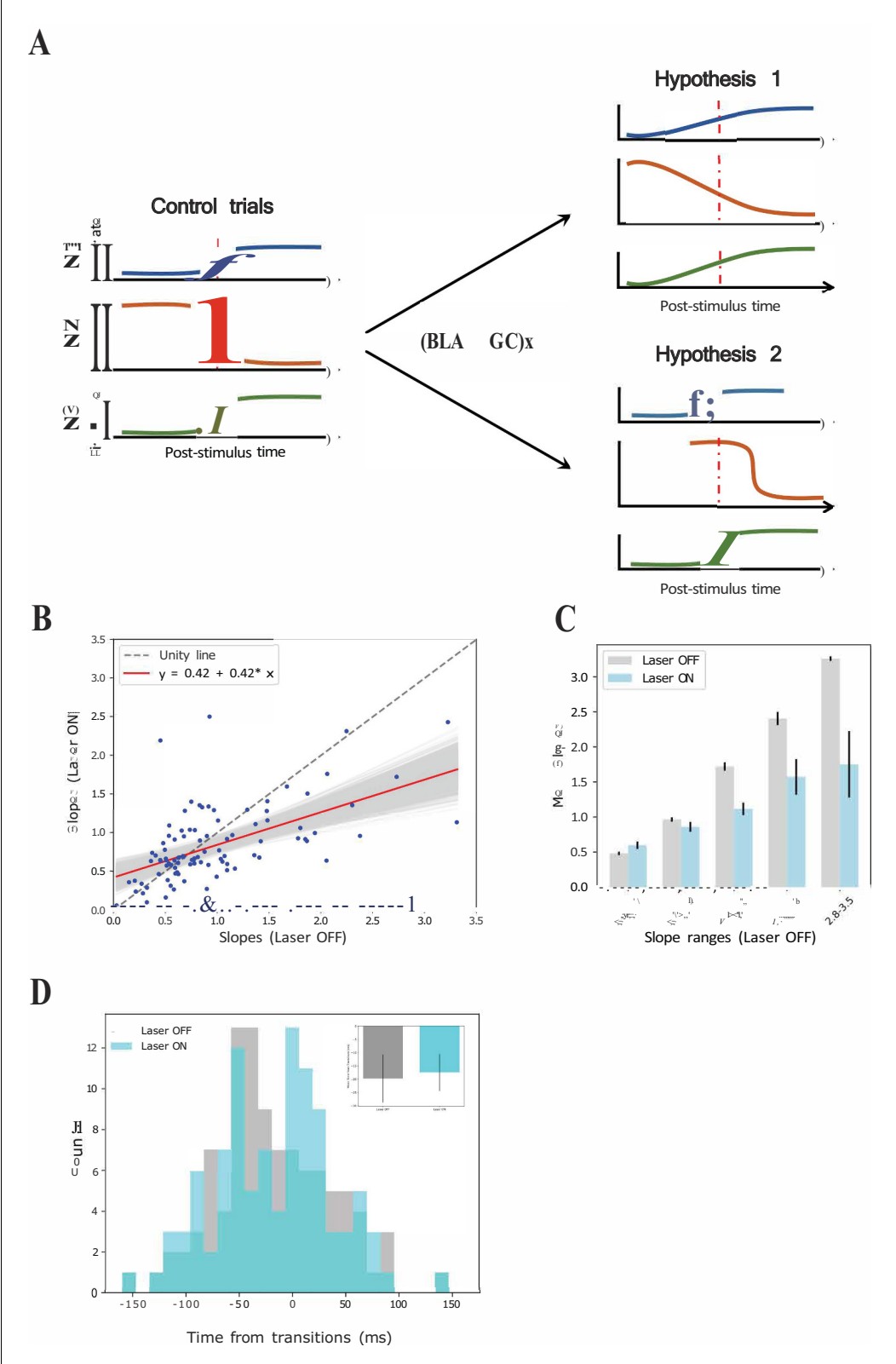

**Figure 10.** GCx reduces the sharpness of firing rate changes around state transitions into the palatability epoch. (**A**) Schematics demonstrating two potential mechanisms by which BLA→GCx can decelerate the rise of palatability correlations. Hypothesis 1: laser stimulation causes a general reduction in the sharpness of firing-rate changes for individual neurons; Hypothesis 2: laser stimulation 'decoupled' the inter-neuron timing of those changes without altering the firing rate dynamics of each individual neuron. Red dashed line in each subpanel indicates the transition time during taste

*Figure 10 continued on next page*

*Figure 10 continued*

responses. (B) Scatter plot shows the slope changes for each GC neuron (Laser-Off [x-axis] against Laser-On [y-axis]). The red line is the regression fit of the data, and its slope was significantly shallower than the unity line (gray-dashed line with slope as 1, that is no impact of laser). (C) Mean slopes (± SEM) of GC neurons that were assigned into five groups based on the slopes in the control, Laser-Off, trials. As revealed, BLA→GCx (light blue bars) significantly reduced the changes in firing rates (slopes) around the state transition time. (D) Histogram of the latencies of when the sharpest slope occurred relative to the transition time across Laser-Off and Laser-On conditions. The mean latency (± SEM) are depicted in the inset, which reveals no significant difference across laser conditions. Accordingly, the reduction in the magnitude of firing changes (i.e. slopes) around the transition time likely accounts for the slowness in palatability correlation ramping up during perturbation on BLA→GC axons.

that the perturbation left unchanged the firing rate dynamics of each individual neuron in the ensemble, but 'de-coupled' these changes across neurons (Figure 10A-bottom).

To test the first hypothesis, we calculated the slopes of each single neuron's firing rate changes across the peri-transition period separately for control and laser trials, and plotted these results in a scatterplot (*Figure 10B*). These data would be expected to hover close to the gray dashed 'unity' (slope = 1) line in this plot if BLA→GCx failed to influence the precipitousness of single-neuron firing rate changes; in fact, however, a regression analysis of the data revealed that slope (0.42) of the fitting line to be significantly shallower than 1 (p<0.05). To probe further, we grouped the data into intervals of slope ranges calculated in control trials (*Figure 10C*); perturbation of activity in BLA→GC axons reduced the rate of firing rate changes (ps < 0.05) across most of these intervals. This pattern of results strongly supports the hypothesis that the firing rate changes of most neurons in an ensemble were 'blurred' in the vicinity of the transition into palatability-related firing when BLA→GC axon activity was perturbed. In other words, disconnecting GC from BLA kept the single neurons within the GC network from changing their firing rates quickly in the vicinity of the transition into palatability-relatedness; this in turn explains the loss of suddenness in the GC ensemble transition into the Palatability state.

We went on to test the second possible mechanism for the ensemble results, asking whether BLA→GCx might have (also) directly disorganized ensembles such that the simultaneity of the transitions was reduced. We identified the times at which each single neuron's firing-rate changes in the vicinity of calculated state transitions reached their maximal slopes (see Materials and methods); these data allow us to determine whether the spread of these times within a simultaneously recorded neural ensemble differed depending on trial type.

*Figure 10D* shows the result of this analysis. While inspection of the figure reveals a good deal of noise in the distributions (likely the result of the small sample), it does not suggest any major differences in the spread of the distribution. Certainly the difference between distributions failed to reach significance ($X^2$ = 13.93, p>0.05), indicating that the coherence of the timing of the GC response dynamic has not been altered by the perturbation of BLA→GC projections, a result consistent with the analysis that found comparable means of the distributions between each trial type (OFF vs ON trials: −17.98 ± 5.29 ms vs. −16.23 ± 5.51 ms). This conclusion is further corroborated by examination, ensemble by ensemble, of the means and standard deviations of the distributions (*Figure 10D* inset), which again appear very similar ($t(6)$ = −0.33, p>0.05). Overall, the loss of the normally observed sharp ensemble transitions into palatability-related firing appears to not reflect decoupling of still sharp single-neuron transitions, but rather an alteration of the basic functioning of the networks, such that entire ensembles of neurons fail to cleanly transition from one state to the next.

The impact of BLA→GCx on slopes changes during state transition times is not universal but epoch-dependent. We repeated the same analysis depicted in *Figure 10A* on firing rate changes comprising transitions into the identity state (which typically occurred 100–600 ms after taste delivery). We found very little impact of BLA→GCx on either the sharpness or timing of these firing-rate changes. *Figure 11A* displays the slopes of this earlier transition for each neuron, with control trials plotted against perturbed trials. As revealed by a regression analysis, the fit line (slope = 0.79; red solid line) did not differ significantly from unity (p>0.05), indicating that the perturbation has little impact on the firing rate changes when the ensemble is transitioning into the identity state. This null effect was confirmed when the range of slopes observed in control trials was divided into subgroups (*Figure 11B*); an ANOVA conducted on these data revealed no significant main effect of Laser ($F_{(1,58)}$ = 1.44, p>0.05) and no significant interaction ($F$ < 1). When examining the timing of sharpest

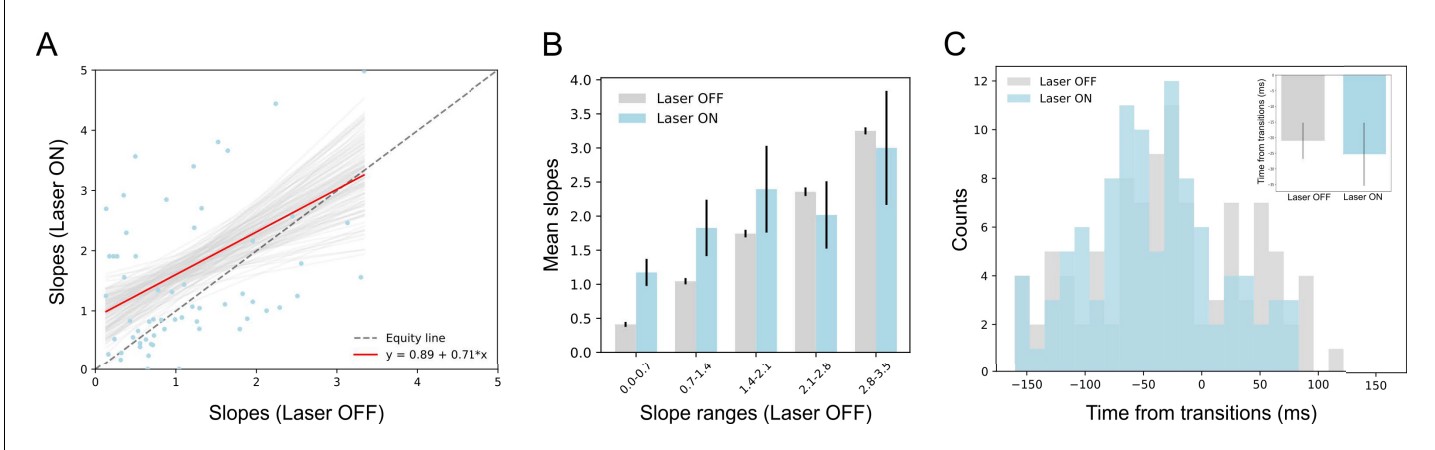

**Figure 11.** GCx has little impact on state transitions into the identity epoch. (**A**) Scatter plot shows the slope changes during the transition times into the identity state (a dominant during 100 ~ 600 ms post-taste delivery) for each GC neuron (Laser-Off [x-axis] against Laser-On [y-axis]). The red line is the regression fit of the data, whose slope is not significantly from the unity line (gray-dashed line with slope as 1, i.e. no impact of laser). (**B**) Mean slopes (± SEM) of GC neurons that were assigned into five subgroups based on the slopes in the control, Laser-Off, trials. Consistent with Panel A, the changes in firing rates (slopes) around the state transition times were comparable between Laser-Off and Laser-On conditions. The seemingly increase in the Laser-On condition for the first two slope ranges (0.0–1.4) is a numerical difference and not supported by statistical significance (see text for more details). (**C**) Histogram of the latencies of when the sharpest slope occurred relative to the transition time across Laser-Off and Laser-On conditions. The mean latencies (± SEM) from each ensemble (the inset) are not significantly different across laser conditions. Thus, BLA→GCx has little impact on the state transitions into the identity epoch.

slope changes relative to transition times (*Figure 11C*), we also found no evidence indicating that BLA→GCx alters the dynamics of firing changes ($X^2$ = 20.06, p>0.05). Overall, this pattern of results further confirms that the input from BLA critically modulates the dynamics of GC taste response by altering palatability-related activity while leaving the identity processing relatively intact.

## Discussion

GC taste processing is not simple. Stimulus responses reflect not just taste identity but also taste palatability, and through the latter GC acts as an essential element in the process of not only 'coding' but also in the decision process for consumption-driven behavior. These different activities are mediated, not by distinct subpopulations of neurons, but by different stages of the response generated by a (mostly) single population of neurons (*Katz et al., 2001*; *Jones et al., 2007*). Such functional complexity all but requires a circuit wherein the dynamically responsive region integrates input from multiple brain areas (*Maffei et al., 2012*; *Staszko et al., 2020*). Given lesion and pharmacological studies demonstrating that GC-governed consummatory behavior—both learned (e.g. CTA) and innate (taste neophobia)—is impaired following BLA dysfunction, it is reasonable to hypothesize that BLA might be a region that vitally interacts with GC during taste processing. The current work tests and confirms this hypothesis, and goes on to characterize that interaction.

The involvement of BLA in GC coding is, as predicted, related to palatability, but the simplest hypothesis—that BLA simply relays palatability-related information to GC—proves too simple. BLA→GCx affected both GC taste identity and taste palatability coding, but only the latter impact was qualitative: while identity-related responses changed, neither the number of responses showing taste-specificity nor our ability to decode taste identity from the responses was altered by laser stimulation; in the absence of BLA inputs, taste-specific information remains readily available in GC. In contrast, the palatability-processing epoch was both quantitatively and qualitatively altered by BLA→GCx—there was a significant loss of palatability-relatedness in GC taste responses, both in terms of number of responses showing significant correlations with palatability and in terms of overall correlation with palatability. This finding extends work demonstrating an impact of whole-region BLA inactivation (*Piette et al., 2012*; see also *Yamamoto et al., 1984*; *Bielavska and Roldan, 1996*): by taking advantage of optogenetics to silence a single axon pathway without silencing

somas in either BLA or GC, we show that it is specifically the direct BLA→GC projection that modulates GC palatability-related activity—conclusions that are consistent with prior findings that acquisition of learned palatability changes enhances the BLA-GC connection (*Grossman et al., 2008*). Optogenetics also made it possible to perform fine-grained, within-session/within-neuron analyses that greatly extended our specific understanding of the function of this projection.

Those more fine-grained analyses allowed us to expose important complexities of the effect, revealing that BLA→GCx, despite being tonic, impacts GC function in an 'epoch-wise' manner (see *Figures 2* and *6*). The onset latencies of that impact were neither random nor exponentially decaying across time: while some responses were impacted starting only a few hundred milliseconds after taste delivery, the distribution of latencies showed peaks around the beginnings of each successive epoch; many responses were altered only during the palatability epoch (i.e., with effect latencies of ~750 ms after taste delivery). Meanwhile, for the vast majority (>90%) of responses impacted at shorter latencies, that impact persisted through the later, palatability epoch. Not only does the impact of BLA→GCx conform to the dynamics that we have reliably observed in GC taste processing (*Katz et al., 2001*; *Jones et al., 2007*; *Sadacca et al., 2012*; *Sadacca et al., 2016*; *Mukherjee et al., 2019*), it also impairs late palatability responses while leaving coding in the identity epoch relatively intact.

Previous studies have reported robust effects of similar perturbations on palatability-guided behavior (CTA learning and taste neophobia; *Gallo et al., 1992*; *Lin et al., 2009*; *Lin and Reilly, 2012*; *Lavi et al., 2018*; *Levitan et al., 2020*). In our hands, however, the impact of BLA→GCx was less complete, and some GC neurons even gained palatability responses in perturbed trials (*Figure 8B*). There are multiple possible explanations for this mild discrepancy. The relatively brief laser stimulation (2.5 s in duration) used here to perturb the system, for instance, could have mitigated the strength of the effect. This explanation seems unlikely, however, given that even briefer stimulation is sufficient to significantly alter the production of orofacial responses evoked by taste presentations (*Mukherjee et al., 2019*). Alternatively, given the fact that our intervention purposefully blocks only the direct projection from BLA to GC while leaving the function of BLA cell bodies intact, residual GC palatability activity could reflect input from BLA routed via a third area that is anatomically connected to both GC and BLA (perhaps lateral hypothalamus [LH]). However, this hypothesis is rendered unlikely by the findings of a previous study (*Piette et al., 2012*) in which BLA cell body inactivation (achieved via muscimol administration): this wholesale BLA manipulation foreshadowed the results presented here—reducing (rather than eliminating) palatability-related information in GC taste responses (and sparing identity coding).

Thus, the fact that palatability-related activity in GC survives removal of BLA input likely means that the hedonic taste information reaches GC via an independent pathway not involving BLA. Two prime candidates are LH and the parabrachial nuclei of the pons (*Yamamoto et al., 1984*; *Kosar et al., 1986*; *Norgren, 1974*): both are directly connected to GC and, importantly, also display similar dynamics of taste responses to those occurring in GC (*Li et al., 2013*; *Baez-Santiago et al., 2016*). Future work will investigate the importance of these regions in the production of GC dynamics.

Regardless of the results of future work, the above-discussed findings, in conjunction with our own analyses, suggest that the central role played by BLA has to do with organizing GC taste response dynamics, rather than with driving palatability-related responses specifically. BLA→GCx significantly 'blurred' the onset of palatability-related activity, which in control trials is a sudden, coherent firing-rate transition. This blurring could potentially account for behavioral deficits in animals with dysfunction in the BLA-GC circuitry, in that the loss of activity synchrony severely reduces the occurrence of learning-related synaptic plasticity (e.g. *Li, 2018*).

While a full explanation of how blocking BLA input causes the incoherent transitions into the palatability epoch in GC must await the results of future experimentation, work from theoretical neuroscience may offer clues to the underlying mechanisms. These studies (e.g. *Jones et al., 2007*; *Miller and Katz, 2010*; *Escola et al., 2011*; *Mazzucato et al., 2015*; *Mazzucato et al., 2019*; *La Camera et al., 2019*) suggest that the taste system functions as a nonlinear 'attractor network,' in which (as we have again shown) taste responses evolve through a sequence of discrete, quasi-stationary 'states,' and that the transitions between these states are jointly determined by the strength of both the attractors and noise impinging upon the network (*Miller and Katz, 2010*). We hypothesize that BLA is an essential part of this dynamical system, and that, given the critical involvement of

BLA in palatability processing, the loss of BLA input may reduce the nonlinearity of the attractor dynamics (and to predispose the network to random noise). Accordingly, the neurons become less well synchronized; they continue to display palatability activity, but in a less coherent manner.

What is the implication of this role of BLA in organizing GC activity? Based on our results, it is reasonable to speculate that during taste processing, BLA actively interacts with GC and coordinates activity among cortical neurons, so that the cortical ensemble can transition suddenly and coherently into the palatability state. An important question, therefore, has to do with when that interaction occurs. *Mukherjee et al., 2019* employed brief (500 ms in duration) optogenetic inhibition of GC itself, shedding light on this issue. When GC was inhibited for the first 500 ms of the taste response, a time period that reliably ended prior to the transition into the palatability epoch, palatability-driven behavior was significantly delayed—a fact that strongly implies that processing intrinsic to GC is important in the time leading up to the Late epoch. Combined with the fact that palatability-related activity occurs much earlier in BLA than it does in GC (*Fontanini et al., 2009*), we speculate that BLA-GC interactions across the first 0.5–1.0 s are responsible for causing the transition into GC palatability epoch; this could be the specific way in which BLA assists/coordinates the processing of this 'emotion-rich' process in GC. Finally, given the nonlinearity of this dynamic population effect, and the dense reciprocal connections between BLA and GC, their interaction is unlikely to be unidirectional, a suggestion that receives support from an earlier study demonstrating that electrical stimulating GC can alter BLA taste responses (*Yamamoto et al., 1984*; also see *Lavi et al., 2018*).

In summary, as revealed in our 20 years of research, taste processing in GC is complex, involving a sequence of firing rate transformations that chart the evolution of those responses from reporting the presence of stimuli on the tongue, to discriminating the taste identity, and then finally to generating affective responses. The nature of this dynamic process almost necessarily requires that GC collaborates with other brain regions, and while such a collaboration could simply involve information passage from one area to another, the results of the current research suggest that GC palatability-related activity is organized by connections to GC from BLA. Future work will assess whether input to GC received from other regions, such as LH, gustatory thalamus (*Cechetto and Saper, 1987*), and parabrachial nucleus play similar or complementary roles in the processing of taste information in the service of modulating feeding behavior.

# Materials and methods

**Key resources table**

| Reagent type (species) or resource | Designation | Source or reference | Identifiers | Additional information |
|---|---|---|---|---|
| Strain, strain background (virus) | AAV9-CAG-ArchT-GFP | UNC Vector Core | Lot #: AV6221E | |
| Antibody | anti-GFP-rabbit IgG (rabbit Polyclonal) | Life Technologies | Cat#: A11122 RRID:AB_221569 | (1:500) |
| Antibody | Alexa Flour 488 donkey anti-rabbit IgG (Donkey Polyclonal) | Life Technologies | Cat#: A21206 RRID:AB_2535792 | (1:200) |
| Strain, strain background (female Long Evans Rat) | | Charles River Laboratories | Strain code: 006 | |
| Chemical compound, drug | Sodium chloride | Fisher Scientific | Cat#: S271-500 | 0.1M (0.29 g/50 ml) |
| Chemical compound, drug | Sucrose | Fisher Scientific | Cat#: S5-500 | 0.3M (5.14 g/50 ml) |

*Continued on next page*

*Continued*

| Reagent type (species) or resource | Designation | Source or reference | Identifiers | Additional information |
|---|---|---|---|---|
| Chemical compound, drug | Citric acid | Fisher Scientific | Cat#: A104-500 | 0.1M (1.05 g/50 ml) |
| Chemical compound, drug | Quinine hydrochloride dihydrate | Sigma-Aldrich | Cat#: Q1125-5G | 1 mM (0.0198 g/50 ml) |
| Software, algorithm | Conda | Conda | RRID:SCR_018317 | |
| Other | DAPI stain | Vector Laboratories, Inc | Cat#: H-1200–10 | |

## Subjects

The experimental subjects were female Long-Evans rats (Charles River Laboratory, Raleigh, NC), singly housed in a vivarium with controlled temperature and 12:12 hr light-dark cycle (lights on at 7:00 am). Given that several previous studies have failed to reveal any significant male/female differences, we chose to use female rats—a decision that maximized the validity of comparisons to our previous papers (many of which have used female rats) and allowed us to take advantage of the fact that female rats are relatively docile to handle (and therefore allow better recording quality than the males). The rats were given ad libitum food and water until experimentation. All procedures complied with the regulations of the Institutional Animal Care and Use Committee (IACUC) at Brandeis University.

## Apparatus

Neural recordings were made in a custom Faraday cage (6 × 24×33 cm) connected to a PC and Raspberry Pi computer (Model 3B). The Pi controlled opening time and duration of solenoid taste delivery valves, and an iris allowing laser stimulation (Laserglow Technologies, Toronto, CA). The PC controlled and saved electrophysiological recordings taken from opto-trode bundles *via* connections to an Intan system (RHD2000 Evaluation System and Amplifier Boards; Intan Technologies, LLC, LA). Each bundle consisted of 32 microwires (0.0015inch formvar-coated nichrome wire; AM Systems) and one optical fiber (0.22 numerical aperture, 200 mm core, inserted through a 2.5 mm multimode stainless-steel ferrule; Thorlabs). The microwire bundle was glued to a custom-made electrode-interface board (San Francisco Circuits) and soldered to a 32-channel Omnetics connector, which was fixed to an adjustable drive (movable along the dorsal-ventral axis) so that multiple recording sessions could be done from a single rat.

## Surgery

Each rat received a pair of surgeries. In the first surgery, rats were anesthetized with an intraperitoneal (ip) ketamine/xylazine mixture (100 mg/kg, 5.2 mg/kg, respectively), and then mounted in a stereotaxic instrument (David Kopf Instruments; Tujunga, CA) with blunt ear bars. A midline incision exposed the skull and a trephine hole (~2 mm diameter) was drilled above BLA in each hemisphere. Thereafter, the construct (AAV-CAG-ArchT-GFP; http://www.med.unc.edu/genetherapy/vectorcore) was infused through a glass pipette (tip ~30 μm) bilaterally into BLA with the following coordinates: Site 1: AP −2.0 mm, ML ±4.9 mm, DV −7.8 mm; Site 2: AP −3.0 mm, ML ±5.1 mm, DV −8.1 mm; all measurements relative to bregma. At each site, 0.5 μl of ArchT virus was infused with a speed of 50 nl/10 s. Approximately 5 min after each infusion, the micropipette was slowly raised out of the brain. After the last infusion, the incision was closed with wound clippers, and the rat was returned to its home cage in the vivarium.

For the second surgery, which took place 3–4 weeks after the first, the skull was again exposed, trephine holes were bored over GC, and multi-channel electrodes + optical fiber ('opto-trode') were implanted just above GC at the coordinates: AP +1.4 mm, ML ±5.0 mm, DV −4.5 mm. Once in place, the opto-trodes were cemented to the skull, along with an intra-oral cannula (IOC), using dental acrylic (*Fontanini and Katz, 2006*).

The rat's body temperature was monitored and maintained at ~37°C by a heating pad throughout the duration of the surgery.

## Experimental design

Following 5 days of recovery from the second surgery, rats were placed on a mild water restriction regimen (25 ml of water offered during the dark portion of the light-cycle). Three days into this schedule, rats began 2 days of habituation to liquid delivered directly to the tongue via IOC, with 120 40 µl infusions of water delivered per session. Thereafter, tastes replaced water, and in vivo electrophysiology recording sessions commenced. All recording sessions took place in the mornings. In each trial during these sessions, one of four gustatory stimuli (0.1M NaCl, 0.3M Sucrose, 0.1M Citric Acid [Acid] and 1 mM Quinine-HCl [QHCl]) was pseudo-randomly chosen for delivery; these stimuli and concentrations were chosen because they ensured, in addition to a range of distinct taste identities, a wide range of palatabilities, thereby facilitating our analyses (see below).

Rats received 30 trials of each taste, each trial consisting of 40 µl infusions; inter-trial intervals were 20 s, which we have found is long enough to allow rats to self-rinse. On 50% of trials for each tastant, activity in BLA→GC axons was perturbed via the opto-trodes; analyses compared perturbation to non-perturbation trials, within-session. Perturbation (nominally inactivation) was induced with a 532 nm (30–40 mW at tip, ArchT) laser, turned on for the 2500 ms following taste delivery.

## Histology

At the completion of the experiment, rats were deeply anesthetized with ketamine/xylazine (120:15 mg/kg, IP) and then perfused transcardially with physiological saline followed by 10% formalin. The brains were extracted and stored in a 10% formalin/30% sucrose solution for at least 3 days, after which they were frozen and sliced on a sliding microtome (Leica SM2010R, Leica Microsystems; thickness 50 µm). Slices were stained and mounted using an established protocol (*Flores et al., 2018*; *Li et al., 2016*), and ArchT-expression in GC and BLA was evaluated *via* inspection of fluorescence (eGFP) under a Keyence fluorescence microscope.

## Neural data collection and analyses

Electrophysiological signals from the micro-electrodes were sampled at 30 kHz using a 32-channel analog-to-digital converter chip (RHD2132) from Intan Technologies. The signals were digitalized online at the head stage and saved to the hard drive of the PC. The collected recordings were then sorted and analyzed off-line with a set of Python analysis scripts (cf. https://github.com/narendramukherjee/blech_clust). Putative single-neuron waveforms (3:1 signal-to-noise ratio) were sorted using a semi-supervised methodology: recorded voltage data were filtered between 300–3000 Hz, grouped into potential clusters by a Gaussian Mixture Model (GMM), and clusters were labeled and/or refined manually (to enhance conservatism) by the experimenters (for details see *Mukherjee et al., 2017*).

## Taste responsivity

A neuron was deemed to be taste responsive if taste-driven firing rates (from 0 to 2 s post-taste delivery) were significantly higher or lower (paired-sample *t-tests*) than pre-stimulus baseline activity (2 s before taste delivery). This analysis collapses across all four tastes, such that 'taste responsivity' indicates purely that a GC neuron responds to taste delivery and reveals nothing about taste specificity (which is described below).

## Taste specificity

To determine whether a GC neuron responds distinctly to some subset of the four taste stimuli, we performed two-way repeated measures analyses of variance (ANOVA), with Taste and Time as variables, on 2 s of post-delivery firing rates (broken into four 500 ms bins, to facilitate comparison with previous reports of taste dynamics; *Katz et al., 2001*). Significance of either the taste main effect or the taste x time Interaction indicates that the firing of the neurons conveys information specific to the taste stimulus—that is, the response to at least one taste differs significantly from the response to at least one other taste.

Note that a neuron may potentially fail to be identified as 'taste responsive' while nonetheless displaying 'taste-specific' activity. This occurs, for example, when some tastes increase GC firing but others decrease it, and is noteworthy because it reflects the multifaceted nature of taste responses in GC.

A jack-knife classification algorithm was also employed to further evaluate the impact of BLA→GCx on how well tastes could be identified (*Foffani and Moxon, 2004*). Single trials of ensemble taste responses taken from 250 to 1750 ms post-stimulus time were first binned into 250 ms bins and compared to the average responses of all other trials for each taste (the single trial being compared was left out/jack-knifed). Using the number of units in each ensemble as the space dimension, Euclidean distance was then calculated from each single trial to the taste template (average responses of each taste). A trial was classified as correct when the minimal distance occurred between the trial and the same taste's template. Performance greater than 25% (i.e. the chance level) indicates taste specificity.

## Taste palatability

Correlation coefficients were calculated to evaluate the degree to which taste-driven firing rates reflect the hedonic value of tastes. As hedonic value can be treated as a ranked variable, we used the nonparametric Spearman's rho test to compute these correlations. A great deal of prior literature (e.g. *Sinclair et al., 2015*; *Tordoff et al., 2015*), including data collected in our laboratory (*Sadacca et al., 2012*; *Li et al., 2013*), confirms that the palatability order of the four tastes used here is reliably sucrose >NaCl > Acid>QHCl.

To reveal the dynamics of palatability-relatedness in GC single-neuron activity, we conducted a 'moving window analysis'—extracting 250 ms segments of each neuron's evoked response to each taste, evaluating the response-palatability correlation, sliding the time window 25 ms forward, performing the analysis again, etc. Responses were deemed palatability-related if this correlation was significant ($p < .05$) for 3 or more consecutive bin windows.

## Determining the impact of BLA→GCx on neuronal firing

We built a hierarchical Poisson generalized linear model (GLM) to estimate the change in taste-evoked GC activity induced by laser stimulation. For each neuron, we specifically compared the mean firing rates during the laser duration (0–2500 ms post-taste delivery) in control and perturbed trials for each taste stimulus. Taking advantage of the Poisson distribution's suitability for spiking data (*Kass and Ventura, 2001*; *Trousdale et al., 2013*), this GLM model can accurately estimate the significance of changes in neural firing. Model parameters include the mean firing rates for every taste and optogenetic perturbation condition, that are in turn composed of taste- and perturbation-specific effects ('random effects') and means across tastes and perturbation conditions ('fixed effects'). For each neuron $n$ in our dataset, we aggregated the spikes produced on trial $i$ of taste $T$ in optogenetic perturbation condition $O$. There were four levels for $T$, corresponding to the four tastes used in our dataset (sucrose, NaCl, Acid, and QHCl). The number of levels for $O$ were two (control and perturbed trials).

We used Markov Chain Monte Carlo methods (MCMC; specifically, the No-U-Turn sampler) to sample the posterior distribution of $firing_{n;T;O}$ for every taste and condition. We performed this analysis for every neuron in our dataset, and ultimately calculated the impact of perturbation on firing as the difference in $firing_{n;T;O}$ between control and perturbation trials. A significant impact of laser stimulation on neuronal firing was concluded if the 95% Bayesian credible interval for these differences in $firing_{n;T;O}$ for a neuron did not overlap 0 (see *Mukherjee et al., 2019* for details).

## Hidden Markov Models (HMMs)

Initially developed for speech recognition, HMM has recently gained attention as a way to analyze in vivo electrophysiology with the utility of determining whether population neuronal activity shifts from ensemble state to ensemble state (*Rabiner, 1989*; *Seidemann et al., 1996*; *Gat et al., 1997*; *Jones et al., 2007*; *Kemere et al., 2008*; *Miller and Katz, 2010*). In accordance with our well-tested model of dynamic GC taste responses, HMM reveals the degree to which data can be described as reflecting a sequence of two taste-specific (first identity- and then palatability-related) states. Trained on neural ensemble data containing neurons from both hemispheres, the algorithm returns its best estimate of the set of underlying states, each defined as a vector of firing rates—one for each neuron—as well as the probability of transitioning from any one state to any other.

### Post-HMM realignment

For each hidden Markov model, we determined the putative underlying state with the highest probability of occurring across all trials within a time window identified, on the basis of current results (*Figure 8*) and previous work (*Katz et al., 2001*; *Grossman et al., 2008*; *Sadacca et al., 2016*), as being the time at which rising ramps of palatability, observed using analyses keyed to stimulus delivery, reach asymptote (between 0.5–1.5 s after taste delivery). These states were deemed the most likely candidate 'palatability' states. The onset time of these 'palatability' states was determined as the time at which the identified state reached the 0.5 probability threshold on each trial. The ensemble data were then re-aligned to these onset times as the 'zero' time point of each trial. Following data alignment, we repeated the above-described palatability analyses that had already been brought to bear on stimulus-aligned data (taste palatability section).

To determine whether BLA→GCx altered the quality of state transitions, we compared how the PSTH changes around the transitions into the late, palatability, state between Laser-Off and Laser-On conditions. Using a moving window analysis (100 ms window, 20 ms step), we measured (1) the slope of PSTH changes around the transition time, and (2) the latency between when the largest PSTH changed and the transition time. The peri-transition time period used for this analysis was 160 ms; the pattern of results, however, remained unchanged if the time-period was limited between 100 and 200 ms.

## Acknowledgements

This work was supported by grants DC006666 (DBK) and DC016706 (J-YL) from the National Institute of Deafness and Other Communication Disorders. For data analysis, we used the Extreme Science and Engineering Discovery Environment (XSEDE), which is supported by National Science Foundation (Grant #: IBN180002). We thank the members of Katz Laboratory for their valuable input.

## Additional information

### Funding

| Funder | Grant reference number | Author |
| --- | --- | --- |
| National Institute on Deafness and Other Communication Disorders | DC006666 | Donald B Katz |
| National Institute on Deafness and Other Communication Disorders | DC016706 | Jian-You Lin |
| National Science Foundation | IBN180002 | Donald B Katz |

The funders had no role in study design, data collection and interpretation, or the decision to submit the work for publication.

### Author contributions

Jian-You Lin, Conceptualization, Software, Formal analysis, Funding acquisition, Investigation, Writing - original draft, Writing - review and editing; Narendra Mukherjee, Conceptualization, Software, Formal analysis, Methodology; Max J Bernstein, Investigation; Donald B Katz, Conceptualization, Resources, Data curation, Supervision, Funding acquisition, Investigation, Visualization, Methodology, Writing - review and editing

### Author ORCIDs

Jian-You Lin (iD) https://orcid.org/0000-0001-7445-2278
Narendra Mukherjee (iD) http://orcid.org/0000-0003-3808-2622
Donald B Katz (iD) https://orcid.org/0000-0002-8444-6063

## Ethics

Animal experimentation: All procedures complied with the regulations of the Institutional Animal Care and Use Committee (IACUC) at Brandeis University (protocol # 19002). The health of the rats were inspected each day following their arrival to the lab. All surgery procedures were conducted under ketamine and xylazine anesthesia, and post-surgical care, to minimize suffering, involved administration of analgesic (meloxicam 0.04 mg/kg) and antibiotic (Pro-Pen-G 150,000 U/kg).

## Decision letter and Author response

Decision letter https://doi.org/10.7554/eLife.65766.sa1
Author response https://doi.org/10.7554/eLife.65766.sa2

# Additional files

## Supplementary files

• Transparent reporting form

## Data availability

Due to the size of the electrophysiology data, the assess of the data can be made available upon request. The code/software used for data analysis can be downloaded from the GitHub repository (https://github.com/narendramukherjee/blech_clust; copy archived at https://archive.softwareheritage.org/swh:1:rev:86d380144b3f85c8951923de873893583bd25edf).

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
