## [Decision Letter]

**Acceptance summary:**

In this study, the authors provide a simple, straightforward, yet quite elegant demonstration of how direct input from BLA supports the temporal evolution of encoding of taste-related information in GC. GC activity was recorded in rats while 4 different tastes were delivered via intraoral cannula; axons in GC from BLA were optogenetically-inactivated for the first 2.5s after taste delivery on half the trials. The authors provide convincing evidence that silencing of BLA->GC input disrupted the evolution of hedonic encoding in GC with relatively minimal effects on detection and sensory identification of tastants. This is a unique, dramatic, and important demonstration of the specific interactions between two areas in processing the full richness of sensory information.

**Decision letter after peer review:**

Thank you for submitting your article "Perturbation of amygdala-cortical projections reduces ensemble coherence of palatability coding in gustatory cortex" for consideration by *eLife*. Your article has been reviewed by 2 peer reviewers, one of whom is a member of our Board of Reviewing Editors, and the evaluation has been overseen by Kate Wassum as the Senior Editor. The reviewers have opted to remain anonymous.

Essential revisions:

On discussion, the reviewers felt that there were only a few essential things to address, and for most the precise outcome of the response was not critical. These are listed below. Additionally the reviewer comments are appended below and we would appreciate if you would also consider and respond to the additional suggestions provided in these reviewers.

1. Please examine the effect of inactivation on identity encoding, perhaps separating the control and inactivated trials into different training sets in the ensemble analysis.

2. Please improve the histology figure.

3. Also please address issues with Figure 6A. This figure shows a number of neurons whose changes in firing rates have an onset within 200ms. This contrasts with the text on page 7 ("optogenetic perturbation of BLA-GC axonal activity, started at the time of taste delivery, seldom impacted the initial 200msec of the GC taste response"). Please resolve this inconsistency.

4. Please include a key resource table.

*Reviewer #1:*

In this study, the authors provide a simple, straightforward, yet quite elegant exploration of the how direct input from BLA supports the temporal evolution of encoding of taste-related information in GC. GC activity was recorded in rats while 4 different tastes were delivered via intraoral cannula; axons in GC from BLA were optogenetically-inactivated for the first 2.5s after taste delivery on half the trials. Activity was analyzed for the well-known 3-epoch evolution of non-specific, identity-based, and finally hedonic information, and the authors provide convincing evidence that silencing of BLA->GC input disrupted hedonic encoding in GC. Notably this was true even though the hedonic epoch extends well beyond the period of inactivation, and the impact of the inactivation of BLA input in GC was relatively specific, being largely limited to the hedonic information, not the detection or identification of the tastants, and especially to the sharpness of its onset and organization. This relatively subtle result is exciting because, while it confirms the general "cartoon" hypothesis regarding BLA function, it also carves out a very specific role for the direct projections in the normal real-time organization of hedonic information. This is a unique, dramatic, and important demonstration of the specific interactions between two areas in processing the full richness of sensory information. Concerns were minor, with one area of potential improvement being and examination of the effect of inactivation on identity encoding.

So overall I think the paper is excellent. I do however have a couple of areas where I have questions or would like more information.

The first is that the authors should spend a little time at the start of the results describing what they did. A few things of particular interest to me would be whether the rats had any experience with the tastants prior to recording, and if not, do analyses of initial sessions show differences from sessions after experience? Also was there any impact on behavior and, relatedly, was inactivation done in every session or is it possible to compare neural measures on control trials within inactivation sessions with measures from trials in sessions without inactivation?

A second area where I would like more information is regarding the claim that BLA inactivation had little or no impact on representation of identity. I appreciate that the numbers of cells responding to identity was not altered, but the authors present convincing evidence that the ensemble changes dramatically. While the authors show that encoding does not differ in a classification analysis, I think this does not fully address the question assuming they are using all trials to train the classifier. I wonder if they might also check whether classification is the same if trials without inactivation are used as the classifier (or vice versa)? If I follow the data, it seems to me that if the analysis is done this way, then it will likely show that the classification performance is better when a withheld trial is from the same set as the training data than when it is not. It seems to me that this will show that even though the identity information is not rendered less detailed or robust by BLA inactivation, it is altered. Assuming this is true, I think it potentially alters or at least provides additional detail for why a subsequent transition to the hedonics might be "blurred".

Finally I wonder if the authors might be more clear in the discussion about the implication of their terminal inactivation and the relatively subtle effect on hedonic information. Specifically while there are dramatic changes to the hedonic information, it is not completely abolished I think? While LH or other areas might be a source of this remaining information, presumably BLA is still active and sending input to other areas. So it may indirectly be still driving the hedonic coding. Is this plausible or does data from ablation or cell body inactivation contradict this? Directly addressing this in the discussion with a paragraph on the significance of the very specific approach would help me understand the authors interpretation.

*Reviewer #2:*

This manuscript from Jian-You Lin and colleagues presents results from experiments investigating the role of the basolateral nucleus of amygdala (BLA) in modulating taste-evoked activity in the gustatory cortex (GC). Specifically, the experiments test the hypothesis that silencing of BLA axons in GC affects spiking dynamics and suppresses coding of taste hedonic value (palatability). The authors rely on electrophysiological recordings and optogenetic silencing of axonal afferents in rats consuming tastants delivered through an intraoral cannula.

Silencing of BLA afferents can lead to either an increase or a reduction in firing rates of GC neurons (with reduction being more prevalent for putative GABAergic interneurons). Prolonged silencing of amygdalar axons leads to changes in GC firing rates that have complex dynamics. Specifically, tonic silencing impacts taste responses at various latencies, generally corresponding with the coding epochs for taste quality and hedonic value. Despite affecting firing rates, BLA afferent silencing has minimal impact on coding of taste quality. Instead, silencing has a significant impact on palatability coding. Finally, the authors use hidden markov model analysis to show that silencing BLA afferents impacts GC metastability.

This is a very interesting manuscript that provides important results on the role of BLA in shaping processing of hedonic value in sensory cortices. The data presented here are a very substantial addition to the literature and will be of wide interest. The article is very well written, the experiments are designed carefully and well-executed. The analyses are thorough, rigorous and provide insights on both single neuron as well as population dynamics. This excellent manuscript could be further improved if the authors could address the following issues:

Figure 1: the author should show higher magnification images detailing axons in GC and cell bodies in BLA. The figures should have a calibration bar. Finally, it would be helpful to have a schematic summarizing optrode placements in GC.

Figure 2: can the authors show raster plots as well? Did the authors observe any bimodal effect, i.e. neurons whose firing rates were first suppressed and then enhanced by optostimulation? It would be helpful to provide a heatmap showing the difference in firing rates (ctrl-opto) for all the neurons whose responses changed with opto silencing.

Page 5: The authors cite Stone at al 2011 in reference to in vitro recordings showing BLA inputs onto pyramidal and GABAergic interneurons. Stone et al. 2011 was an in vivo study. They may want to consider also citing Haley et al. 2016 and 2020 (both are in vitro studies).

Figure 6A shows a number of neurons whose changes in firing rates have an onset within 200ms. This contrast with the text on page 7 ("optogenetic perturbation of BLA-GC axonal activity, started at the time of taste delivery, seldom impacted the initial 200msec of the GC taste response"). The authors should consider downplaying their statement.

Figure 7D: the author should include in the panel the classification performance averaged across all tastants.

Figure 9: is the effect of BLA silencing on state transition time restricted to the palatability epoch? Is the transition time between first and second state (conceivably happening in the first 500 ms) also dilated by BLA axonal silencing? It would be informative to add this analysis and assess the specificity of the effect.

How many rats were used for this study? The authors should justify why only females.

The results from the HMM analysis are extremely interesting, as they suggest that BLA inputs may modulate attractor dynamics and metastability. The authors should consider elaborating a bit on the implications of this point in the discussion.

---

## [Author Response]

Essential revisions:On discussion, the reviewers felt that there were only a few essential things to address, and for most the precise outcome of the response was not critical. These are listed below. Additionally the reviewer comments are appended below and we would appreciate if you would also consider and respond to the additional suggestions provided in these reviewers.1. Please examine the effect of inactivation on identity encoding, perhaps separating the control and inactivated trials into different training sets in the ensemble analysis.

This analysis has been added to the Results section, as detailed in the response to Reviewer 1.

2. Please improve the histology figure.

The figure has been updated with higher magnification pictures.

3. Also please address issues with Figure 6A. This figure shows a number of neurons whose changes in firing rates have an onset within 200ms. This contrasts with the text on page 7 ("optogenetic perturbation of BLA-GC axonal activity, started at the time of taste delivery, seldom impacted the initial 200msec of the GC taste response"). Please resolve this inconsistency.

The sentence has been revised, and the inconsistency resolved (see response to Reviewer 2).

4. Please include a key resource table.

A resource table is submitted with the revision.

Reviewer #1:In this study, the authors provide a simple, straightforward, yet quite elegant exploration of the how direct input from BLA supports the temporal evolution of encoding of taste-related information in GC. GC activity was recorded in rats while 4 different tastes were delivered via intraoral cannula; axons in GC from BLA were optogenetically-inactivated for the first 2.5s after taste delivery on half the trials. Activity was analyzed for the well-known 3-epoch evolution of non-specific, identity-based, and finally hedonic information, and the authors provide convincing evidence that silencing of BLA->GC input disrupted hedonic encoding in GC. Notably this was true even though the hedonic epoch extends well beyond the period of inactivation, and the impact of the inactivation of BLA input in GC was relatively specific, being largely limited to the hedonic information, not the detection or identification of the tastants, and especially to the sharpness of its onset and organization. This relatively subtle result is exciting because, while it confirms the general "cartoon" hypothesis regarding BLA function, it also carves out a very specific role for the direct projections in the normal real-time organization of hedonic information. This is a unique, dramatic, and important demonstration of the specific interactions between two areas in processing the full richness of sensory information. Concerns were minor, with one area of potential improvement being and examination of the effect of inactivation on identity encoding.So overall I think the paper is excellent. I do however have a couple of areas where I have questions or would like more information.The first is that the authors should spend a little time at the start of the results describing what they did. A few things of particular interest to me would be whether the rats had any experience with the tastants prior to recording, and if not, do analyses of initial sessions show differences from sessions after experience? Also was there any impact on behavior and, relatedly, was inactivation done in every session or is it possible to compare neural measures on control trials within inactivation sessions with measures from trials in sessions without inactivation?

1. The result section has been revised to begin with a brief description of the experimental procedures, noting (among other things) that our rats arrived at recording/optogenetics sessions essentially taste-naïve.

2. We have conducted (and added to the manuscript) a comparison of the impact of BLA→GCx on initial and later recording sessions. The results of this analysis reveal little in the way of significant between-session differences in this impact (lines 139-147).

3. We now explicitly note that we chose to take advantage of the felicitous properties of optogenetics, restricting ourselves to performing within-session analyses comparing trials in which the BLA→GC connection was perturbed and those without perturbation for purposes of experimental control. Future work will introduce testing of between-session and behavioral hypotheses!

A second area where I would like more information is regarding the claim that BLA inactivation had little or no impact on representation of identity. I appreciate that the numbers of cells responding to identity was not altered, but the authors present convincing evidence that the ensemble changes dramatically. While the authors show that encoding does not differ in a classification analysis, I think this does not fully address the question assuming they are using all trials to train the classifier. I wonder if they might also check whether classification is the same if trials without inactivation are used as the classifier (or vice versa)? If I follow the data, it seems to me that if the analysis is done this way, then it will likely show that the classification performance is better when a withheld trial is from the same set as the training data than when it is not. It seems to me that this will show that even though the identity information is not rendered less detailed or robust by BLA inactivation, it is altered. Assuming this is true, I think it potentially alters or at least provides additional detail for why a subsequent transition to the hedonics might be "blurred".

The reviewer is absolutely right: while GC responses remain discriminative in the absence of BLA input, the specifics of taste coding are still likely changed by the removal of BLA input. We have performed the new analysis suggested, evaluating how well a classifier trained on non-perturbed trials classifies tastes delivered in perturbed trials (and vice versa). The results (Figure 7E) show, as the reviewer predicted, that the classification is significantly worse when evaluated across conditions than when within each laser condition – particularly when the sample was limited to neurons for which activity was altered by BLA→GCx. These results suggest, as we now note in the manuscript, that in the absence of BLA input, GC does not loss its capacity to process taste identity, but does code identity differently. This fact is particularly intriguing given the fact that further analysis (described below) suggests that the impact of BLA→GCx on the transition into palatability-related firing is NOT paralleled by analogous earlier changes to the transition into identity coding.

Finally I wonder if the authors might be more clear in the discussion about the implication of their terminal inactivation and the relatively subtle effect on hedonic information. Specifically while there are dramatic changes to the hedonic information, it is not completely abolished I think? While LH or other areas might be a source of this remaining information, presumably BLA is still active and sending input to other areas. So it may indirectly be still driving the hedonic coding. Is this plausible or does data from ablation or cell body inactivation contradict this? Directly addressing this in the discussion with a paragraph on the significance of the very specific approach would help me understand the authors interpretation.

The reviewer is correct that BLA→GCx does not abolish GC palatability activity – a fact that is theoretically important, and that does, as the reviewer suggests, raise the possibility that BLA could still be modulating GC activity via other brain regions when BLA→GC axons are silenced. We consider this explanation to be unlikely, however, given a prior study (Piette et al., 2012) demonstrating that wholesale, semi-chronic BLA cell body inactivation (achieved via muscimol administration) causes a similar reduction in GC palatability activity. We therefore suspect (and note in the manuscript) that the significant residual palatability activity reflects processing performed in regions other than BLA itself, for example the lateral hypothalamus (Li et al., 2013) or parabrachial nuclei of the pons (Baez-Santiago et al., 2016). More detailed speculation is now found in the Discussion (lines 598-614) – specifically, a proposal that evolutionarily-important palatability processing is performed redundantly across the taste neuroaxis, and that the primary “job” of BLA→GC cooperation might be to dynamically organize this processing.

Reviewer #2:[…] This excellent manuscript could be further improved if the authors could address the following issues:Figure 1: the author should show higher magnification images detailing axons in GC and cell bodies in BLA. The figures should have a calibration bar. Finally, it would be helpful to have a schematic summarizing optrode placements in GC.

An updated figure is provided that combines a higher-resolution photomicrograph with a schematic showing the placements of all optrodes.

Figure 2: can the authors show raster plots as well? Did the authors observe any bimodal effect, i.e. neurons whose firing rates were first suppressed and then enhanced by optostimulation? It would be helpful to provide a heatmap showing the difference in firing rates (ctrl-opto) for all the neurons whose responses changed with opto silencing.

Ooh, this was a good question, to which we simply did not know the answer. We have added raster plots to PSTH-bearing Figure 2, and have added a panel and analysis to Figure 2 that reveals that the direction of response change was almost always monolithic for individual responses (Figure 2E).

Page 5: The authors cite Stone at al 2011 in reference to in vitro recordings showing BLA inputs onto pyramidal and GABAergic interneurons. Stone et al. 2011 was an in vivo study. They may want to consider also citing Haley et al. 2016 and 2020 (both are in vitro studies).

We apologize for these mistakes in referencing. The references have been appropriately updated in the revised manuscript.

Figure 6A shows a number of neurons whose changes in firing rates have an onset within 200ms. This contrast with the text on page 7 ("optogenetic perturbation of BLA-GC axonal activity, started at the time of taste delivery, seldom impacted the initial 200msec of the GC taste response"). The authors should consider downplaying their statement.

As suggested, the statement has been revised and now it reads as the following: "… while the laser was turned on at the time of taste delivery, in the majority of cases optogenetic perturbation of BLA-GC axonal activity impacted GC taste response only after substantial delays, leaving the initial 200msec of the responses unaltered. "

Figure 7D: the author should include in the panel the classification performance averaged across all tastants.

As suggested, the panel is now included (Figure 7E).

Figure 9: is the effect of BLA silencing on state transition time restricted to the palatability epoch? Is the transition time between first and second state (conceivably happening in the first 500 ms) also dilated by BLA axonal silencing? It would be informative to add this analysis and assess the specificity of the effect.

This is another good idea, and we now provide an analysis of the speed of transition into the state that dominates responses prior to the transition into the palatability-related state (i.e., during the period of 100-600 ms post-taste delivery, the putative “identity state”), evaluated in the same way as in Figure 10. In contrast to its impact on transitions of the palatability epoch, BLA→GCx was shown to have little impact on the dynamics of firing-rate changes around the transition times into the identity epoch (see Figure 11).

How many rats were used for this study? The authors should justify why only females.

We now provide the requested information in lines 133-135 and 678-682. Given that several previous studies have not revealed any significant M/F differences, we chose to use female rats – a decision that maximized the validity of comparisons to our previous papers (many of which have used female rats) and allowed us to take advantage of the fact that female Long Evans rats are relatively docile to handle (and therefore allow better recording quality than the males).

The results from the HMM analysis are extremely interesting, as they suggest that BLA inputs may modulate attractor dynamics and metastability. The authors should consider elaborating a bit on the implications of this point in the discussion.

We appreciate the opportunity to do precisely this – please see lines 625-638. The text now reads as follows:

“While a full explanation of how blocking BLA input causes the incoherent transitions into the palatability epoch in GC must await the results of future experimentation, work from theoretical neuroscience may offer clues to the underlying mechanisms. […] Accordingly, the neurons become less well synchronized; they continue to display palatability activity, but in a less coherent manner.”